# A Survey on Quantum Machine Learning: Basics, Current Trends, Challenges, Opportunities, and the Road Ahead

## Abstract

Quantum Computing (QC) claims to improve the efficiency of solving complex problems, compared to classical computing. When QC is integrated with Machine Learning (ML), it creates a Quantum Machine Learning (QML) system. This paper aims to provide a thorough understanding of the foundational concepts of QC and its notable advantages over classical computing. Following this, we delve into the key aspects of QML in a detailed and comprehensive manner.

In this survey, we investigate a variety of QML algorithms, discussing their applicability across different domains. We examine quantum datasets, highlighting their unique characteristics and advantages. The survey also covers the current state of hardware technologies, providing insights into the latest advancements and their implications for QML. Additionally, we review the software tools and simulators available for QML development, discussing their features and usability.

Furthermore, we explore practical applications of QML, illustrating how it can be leveraged to solve real-world problems more efficiently than classical ML methods. This survey aims to consolidate the current landscape of QML and outline key opportunities and challenges for future research.

## 1 Introduction

Machine Learning (ML) systems are well-established tools for identifying patterns in data and generalizing complex, nonlinear problems. These systems have found applications in various domains, including computer vision, healthcare, finance, and the automotive industry. However, ML practitioners must navigate the substantial computing resources required to run large ML models. Despite employing numerous hardware-aware optimizations such as compression and approximations (Marchisio et al., 2018; Hanif et al., 2018; Marchisio et al., 2019; Hanif et al., 2022; Leon et al., 2023), the limitations of current computing infrastructures and technologies constrain the computational capabilities of ML systems.

The high computational demands of modern ML models necessitate advanced hardware development. According to Moore's law, the number of transistors on an integrated circuit doubles approximately every two years (Gustafson, 2011). However, recent analyses indicate that while Moore's Law has historically driven exponential growth in transistor density, this trend has decelerated in recent years, prompting discussions about its continued applicability in the face of physical and economic challenges (Zhang, 2022). This physical saturation restricts computational power, leading to delays in processing, developmental, and scientific discovery within the ML community. For instance, training Large Language Models with hundreds of billions of parameters and trillions of tokens is extremely compute-intensive (OpenAI, 2023). Such tasks require massive investments in time and hardware resources, which only a few high-end companies can afford. To overcome these physical limits and support further discoveries, there is an urgent need to explore new technological avenues of hardware systems that can enhance the computational efficiency for solving given real-world problems by closely simulating and comprehending them.

One of the most promising solutions to this bottleneck is Quantum Computing (QC). Initially proposed by Feynmann (1982) and further developed by Preskill (2018; 2023), QC systems exploit quantum mechanical

Table 1: Qualitative comparison between our work and related QML surveys.

| QML Survey | Year | # Pages | # Refs. | QC Details | QC Advantages | QML Algs. | Datasets/Encoding | HW Techs. | SW Tools | Applications |
|---|---|---|---|---|---|---|---|---|---|---|
| (Aïmeur et al., 2006) | 2006 | 12 | 24 | ✓ | ✗ | ✓ | ✓ | ✗ | ✗ | ✗ |
| (Schuld et al., 2014) | 2014 | 19 | 71 | ✗ | ✓ | ✓ | ✗ | ✗ | ✗ | ✗ |
| (Adcock et al., 2015) | 2015 | 38 | 97 | ✗ | ✗ | ✓ | ✓ | ✗ | ✗ | ✓ |
| (Montanaro, 2016) | 2016 | 17 | 100 | ✗ | ✓ | ✓ | ✗ | ✓ | ✗ | ✓ |
| (Dunjko & Briegel, 2017) | 2017 | 106 | 309 | ✓ | ✓ | ✓ | ✓ | ✗ | ✗ | ✓ |
| (Ciliberto et al., 2018) | 2018 | 33 | 187 | ✓ | ✓ | ✓ | ✗ | ✗ | ✗ | ✓ |
| (Jeswal & Chakraverty, 2018) | 2019 | 15 | 74 | ✓ | ✗ | ✓ | ✗ | ✗ | ✗ | ✓ |
| (Resch & Karpuzcu, 2019) | 2019 | 29 | 309 | ✓ | ✓ | ✓ | ✗ | ✓ | ✗ | ✓ |
| (Ramezani et al., 2020) | 2020 | 8 | 71 | ✓ | ✗ | ✓ | ✗ | ✗ | ✗ | ✗ |
| (Chakraborty et al., 2020) | 2020 | 6 | 47 | ✗ | ✗ | ✓ | ✗ | ✗ | ✗ | ✓ |
| (Bharti et al., 2020) | 2020 | 17 | 165 | ✓ | ✗ | ✓ | ✗ | ✗ | ✗ | ✓ |
| (Mangini et al., 2021) | 2021 | 9 | 75 | ✗ | ✓ | ✓ | ✓ | ✗ | ✗ | ✗ |
| (Kwak et al., 2021) | 2021 | 4 | 41 | ✓ | ✗ | ✓ | ✗ | ✗ | ✗ | ✓ |
| (Zhao & Wang, 2021) | 2021 | 14 | 73 | ✗ | ✓ | ✓ | ✗ | ✗ | ✗ | ✗ |
| (N et al., 2022) | 2022 | 6 | 21 | ✗ | ✗ | ✓ | ✗ | ✗ | ✗ | ✗ |
| (Umer & Sharif, 2022) | 2022 | 17 | 154 | ✓ | ✗ | ✓ | ✗ | ✗ | ✗ | ✓ |
| (Schetakis et al., 2022) | 2022 | 12 | 59 | ✗ | ✗ | ✓ | ✗ | ✗ | ✓ | ✗ |
| (Markidis, 2023) | 2023 | 19 | 116 | ✗ | ✗ | ✓ | ✓ | ✓ | ✓ | ✓ |
| (Massoli et al., 2023) | 2023 | 49 | 199 | ✓ | ✓ | ✓ | ✓ | ✗ | ✗ | ✓ |
| (Dalzell et al., 2023) | 2023 | 337 | 851 | ✓ | ✓ | ✓ | ✓ | ✗ | ✗ | ✓ |
| (Huynh et al., 2023) | 2023 | 59 | 301 | ✗ | ✓ | ✓ | ✓ | ✗ | ✓ | ✓ |
| (Tychola et al., 2023) | 2023 | 21 | 87 | ✓ | ✓ | ✓ | ✓ | ✗ | ✗ | ✓ |
| (Evans et al., 2024) | 2024 | 44 | 42 | ✓ | ✓ | ✓ | ✓ | ✗ | ✗ | ✗ |
| (Wang & Liu, 2024) | 2024 | 30 | 165 | ✗ | ✗ | ✓ | ✗ | ✗ | ✓ | ✗ |
| (Chen et al., 2024) | 2024 | 44 | 134 | ✓ | ✓ | ✓ | ✓ | ✗ | ✗ | ✓ |
| Our work | 2025 | 62 | 235 | ✓ | ✓ | ✓ | ✓ | ✓ | ✓ | ✓ |

phenomena to significantly improve performance and information processing compared to classical systems. Unlike classical systems that struggle to capture the complexity of natural processes, quantum computers operate on similar principles to those found in nature. This similarity suggests that QC could help us better understand natural phenomena. Moreover, QC has the potential to solve problems more cost-efficiently by enabling advanced optimizations and reducing computation time.

The Quantum Machine Learning (QML) paradigm represents an excellent opportunity for researchers and industries to achieve remarkable discoveries and design efficient solutions for complex real-world problems. QML extends classical ML by leveraging quantum effects, such as superposition and entanglement, to process data in exponentially large feature spaces (Lloyd et al., 2013; Biamonte et al., 2017; Schuld et al., 2014). QML systems, driven towards practicality and improved performance over classical systems, open new avenues for the community to discover, build, and align their designs across different levels of the quantum stack (Schuld & Killoran, 2022). This integration of QC and ML could lead to groundbreaking advancements and a deeper understanding of the world around us.

## 1.1 Scope and Contributions

In this work, we explore the extent to which the overlap between ML and QC has been investigated and identify the future potential of this intersection. This paper presents a systematic and critical discussion of the current status and future perspectives in the field of QML. Our aim is to provide readers with a comprehensive understanding of QML by detailing the currently available algorithms, frameworks, technologies, and tools.

The primary goal of this survey is to introduce and provide substantial knowledge on QML, laying a solid foundation for solving new and advanced problems. In contrast to prior QML surveys (Aïmeur et al., 2006; Schuld et al., 2014; Adcock et al., 2015; Montanaro, 2016; Dunjko & Briegel, 2017; Ciliberto et al., 2018; Jeswal & Chakraverty, 2018; Resch & Karpuzcu, 2019; Ramezani et al., 2020; Chakraborty et al., 2020; Bharti et al., 2020; Mangini et al., 2021; Kwak et al., 2021; Zhao & Wang, 2021; N et al., 2022; Umer & Sharif, 2022; Schetakis et al., 2022; Markidis, 2023; Massoli et al., 2023; Dalzell et al., 2023; Huynh et al., 2023; Tychola et al., 2023; Evans et al., 2024; Wang & Liu, 2024; Chen et al., 2024), we provide a well-structured and thoroughly comprehensive collection of information and concepts that summarize the current state of

development of QML. Table 1 illustrates the comparison between this work and a selection of related QML surveys, highlighting the features and topics each covers. It becomes evident from the table that previous surveys have at least one missing feature, whereas this work delivers a detailed review encompassing all key aspects of QML. **The contributions of this paper are summarized below.**

- **Detailed Overview of Quantum Computers:** We present an in-depth analysis of quantum computers, addressing current challenges, existing techniques, and ongoing efforts to develop viable solutions.

- **Review of State-of-the-Art QML Algorithms:** We review the latest QML algorithms, encoding techniques, datasets, hardware technologies, software tools, and their applications, providing a comprehensive understanding of the field's current capabilities.

- **Critical Discussion of Open Research Challenges:** We offer a critical examination of the open research challenges and future directions in the QML field, discussing its practical potential and the steps needed to advance the technology further.

By encompassing these areas, this paper serves as a valuable resource for researchers and practitioners, offering a thorough overview of the current landscape and future possibilities in QML.

### 1.2  Outline

This paper is organized into different sections and sub-sections. Section 1 introduces the problem, the scope, and the contributions of this paper. Section 2 presents foundational theoretical concepts of general quantum computers, the challenges of the current technologies, and the state-of-the-art methodologies to overcome these issues. Section 3 provides an overview of the claimed and proven advantages of QC compared to classical computing. This section discusses the motivations that are driving researchers and industries to design QC and QML systems. Section 4 discusses the plethora of the most common QML algorithms that are present in the literature and provides an overview of their applicability. Section 5 provides the necessary information to understand the existing methods for manipulating quantum data in a way that it can be used in the QML workflow, along with a curated list of dataset resources from the QML perspective. Section 6 introduces the existing tools and technologies that are available for QML practitioners to experiment with and investigate further. Section 7 presents the current and potential applications of QML that majorly highlight their benefits over classical computing. Section 8 concludes the paper and provides future outlooks in the QML field.

## 2  Quantum Computing Preliminaries

This section provides an overview of the fundamental concepts in the field of quantum systems. It explains the core features and characteristics of QC systems to build a strong conceptual model of quantum computers, with necessary details for readers to understand the rest of the paper.

### 2.1  Understanding the Qubit

The bit forms the fundamental processing unit of a classical computing system. As shown in Figure 1a, it can hold one of the two mutually exclusive values, i.e., either 0 or 1 at one instance of time, representing classical computation and classical information. On the other hand, a quantum computer's fundamental unit for quantum computation and quantum information is a *quantum bit* or *qubit*, which can exist in a continuum of states (basis states).

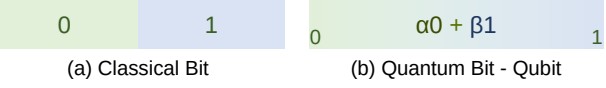

(a) Classical Bit          (b) Quantum Bit - Qubit

Figure 1: **(a)** Classical bit vs. **(b)** Quantum Bit with $\alpha$ and $\beta$ as respective amplitude values of a state-vector, creating a superposition state where $|\alpha|^2 + |\beta|^2 = 1$ as per the Max-Born Rule, satisfying the completeness equation which states to have all probabilities summing up to 1 (Nielsen & Chuang, 2010b). $|\alpha|^2$ and $|\beta|^2$ give the respective probability for each state.

Understanding the qubit as an abstract mathematical object is crucial for the development of quantum circuits. Although the physical realization of qubits is important, the fundamental properties and specifications of qubits are dictated by quantum mechanical principles and remain consistent regardless of their physical form. Unlike a classical bit, which can only be in one of two states (0 or 1) at any given time, a qubit can exist in a superposition of both 0 and 1 simultaneously (see Figure 1b).

Mathematically, a qubit is represented as a unit vector in a two-dimensional complex Hilbert space (see Equation (1)).

$$|\psi\rangle = \alpha |0\rangle + \beta |1\rangle, \quad \text{where } \alpha, \beta \in \mathbb{C}, \quad |\alpha|^2 + |\beta|^2 = 1 \tag{1}$$

Here, $|0\rangle$ and $|1\rangle$ are the computational basis states, and the coefficients $\alpha$ and $\beta$ are complex probability amplitudes. The condition $|\alpha|^2 + |\beta|^2 = 1$ ensures normalization, according to the Max-Born rule.

This quantum mechanical property of representing any qubit state as a linear combination of basis states, called *superposition*, allows a qubit to embody a range of probabilities for being in either state, reflecting its inherent quantum nature. This abstract representation of a qubit is consistent across any physical realization of the qubit, whether it is implemented using superconducting circuits, trapped ions, or any other technology.

On the other hand, the resultant state of the quantum computation system's processing is dictated by the probabilistic occurrence of the collective system's states as a solution over multiple experiment sample runs (called *shots*). For each shot, the resulting solution over the *super-positioned* states, at the final computational stage of the system circuit after state manipulation (quantum processing) using a set of gates, passes through the measurement operation to form the measured state. It is represented through the basis states $|0\rangle$ and $|1\rangle$, followed by the final probabilistic distribution over possible measured state outcomes as solutions to the problem (Marinescu & Marinescu, 2012a).

As shown in Figure 2, the *Bloch sphere* provides a geometric representation of a qubit's state, where any pure qubit state corresponds to a point on the surface of the sphere. This visualization aids in understanding qubit manipulations and quantum gate operations.

## 2.2 Superposition

The *superposition* principle is one of the most intriguing and counterintuitive aspects of quantum mechanics. It states that a quantum system can exist in multiple states simultaneously until it is measured. The space in which these states exist is called *Hilbert space*, a mathematical framework that allows the description of quantum states as vectors. The principle of superposition allows a qubit to exist simultaneously in multiple states until measured. Upon measurement, the qubit collapses to one of the basis states, $|0\rangle$ or $|1\rangle$, with probabilities determined by the squared magnitudes of the respective amplitudes. For example, a qubit in the state $|\psi\rangle = \frac{1}{\sqrt{2}}(|0\rangle + |1\rangle)$ has equal probabilities of being measured in either basis state.

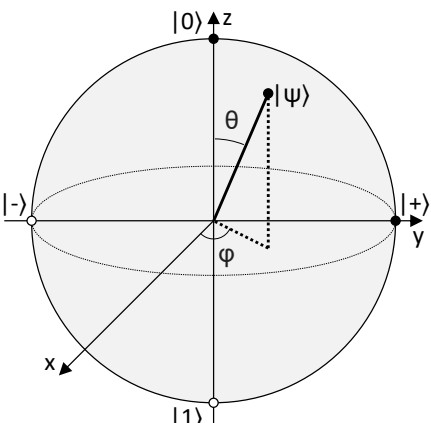

Figure 2: Bloch sphere for a single two-level qubit system.

In the context of qubits, superposition means that a qubit can represent both 0 and 1 simultaneously, allowing quantum computers to process a vast amount of information in parallel. This property is what gives quantum computers their potential to solve certain complex problems much more efficiently than classical computers.

While the physical realization of qubits is key for the development of quantum circuits, it is more important to understand qubits as abstract mathematical objects. These objects adhere to quantum mechanical principles, which remain consistent regardless of the physical form used for qubit realization. A qubit can be mathematically described as a linear combination (or superposition) of its basis states, $|0\rangle$ and $|1\rangle$. This abstract representation is crucial for designing and understanding quantum algorithms and systems, as it encapsulates the fundamental quantum properties such as superposition and entanglement.

The wave-particle duality and the Max-Born Rule are foundational concepts that lead to a deeper understanding of the superposition principle. The superposition principle, in turn, is essential for grasping the unique capabilities of qubits, which are the building blocks of quantum computing. Understanding these principles not only clarifies the theoretical underpinnings of quantum mechanics but also informs the practical development of quantum technologies.

### 2.2.1 Single Qubit Superposition

As illustrated in Figure 3, the state of the qubit is composed of a mixed presence of state $|0\rangle$ and state $|1\rangle$, i.e., with superposition, in which the amplitude values can be obtained after measurement. For each independent quantum system, the probabilities associated with the measured states are illustrated as separate blocks. This behavior is dictated quantitatively by the amplitude value in context to the wave function driven by the underlying quantum mechanics of the system. Using the Max-Born rule, we sum the squared norm of all amplitudes of the superposition states, which in our example are $\alpha$ and $\beta$. Converting the states into probabilities makes the system more intuitive and interpretable.

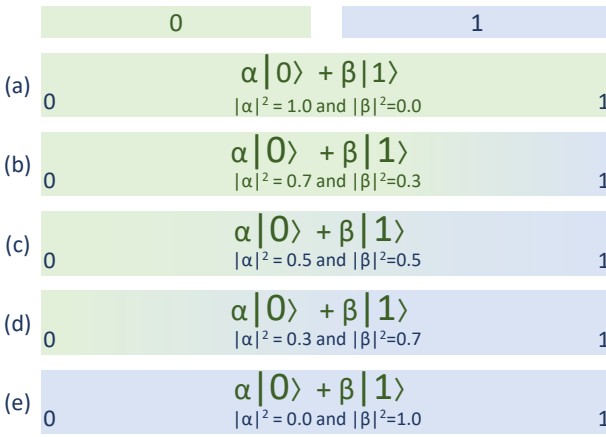

*Quantum physics uses state-vectors to describe the state of the system.* Figure 3 illustrates various superposition states possible for a single qubit system resulting in probabilities. Currently, there are 5 different arbitrarily chosen superposition states shown for a single qubit, where each block is an independent qubit system with its own corresponding state-vector containing amplitude values $\alpha$ and $\beta$. We can observe that the amplitudes of $\alpha$ and $\beta$ can hold any value from the complex number space defining the quantum mechanical system. However, for practical use of holding and processing information, some rules apply to the amplitude values in the state-vector. According to the Max-Born rule, all amplitude values in the state-

Figure 3: A few examples for single qubit Superposition cases. Each block represents an independent quantum system visualizing the resultant measured state of the system with gradient depicting probabilities computed from amplitude values $\alpha$ and $\beta$ of respective superpositions. A superposition of a single qubit **(a)** fully measured to be state $|0\rangle$. **(b)** with 70% probability to be measured in state $|0\rangle$ and 30% for state $|1\rangle$. **(c)** Equal probability to be measured in state $|0\rangle$ and state $|1\rangle$. **(d)** with 30% probability to be measured in state $|0\rangle$ and 70% for state $|1\rangle$. **(e)** Fully measured to be in state $|1\rangle$.

vector are such that the sum of their squared norm is always equal to 1. In other words, we observe from the Max-Born rule that amplitudes and probabilities are related to each other. To use the probabilities, we should satisfy the property that their sum is equal to 1. To meet these conditions, the amplitude values of the state-vector must be normalized.

### 2.2.2 Two-Qubit Superposition

Let us now take one step forward and visualize the superposition samples for a two-qubit system. In Figure 4 and Figure 5, all the principles involved in the system are the same as those applied in a single qubit. Still, now the quantum system can exist in 4 (in general, $2^n$ for an n-qubit system) superpositioned states with 2 qubits at once in the system with respective amplitudes for each state until the measurement operation is performed to determine the solution state. Each figure shows one independent quantum system with various examples of superpositions to elaborate the idea. Equation (2) also illustrates that every state has a corresponding amplitude value ($\alpha$, $\beta$, $\gamma$, $\eta$), whose squared value corresponds to the probability of that state being the measured state.

$$system\ state = \alpha\,|00\rangle + \beta\,|01\rangle + \gamma\,|10\rangle + \eta\,|11\rangle \tag{2}$$

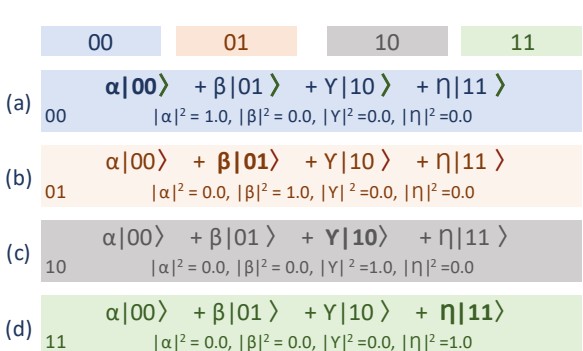

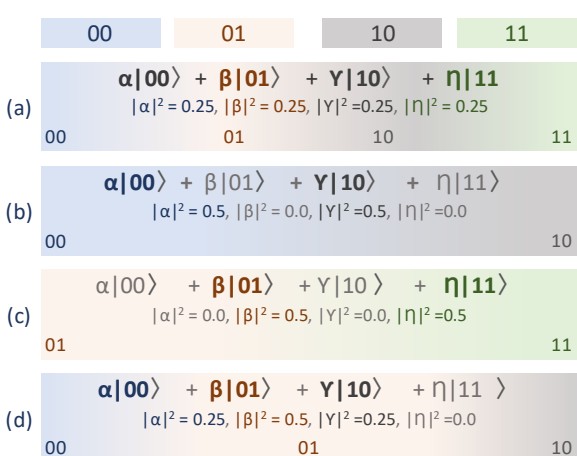

Figure 4: A few examples of two-qubit superposition cases depicting measured outcomes. **(a)** State $|00\rangle$ as measured state with 100% probability. **(b)** State $|01\rangle$ as measured state with 100% probability. **(c)** State $|10\rangle$ as measured state with 100% probability. **(d)** State $|11\rangle$ as measured state with 100% probability.

Figure 5: Other examples of two-qubit superposition cases. **(a)** 25% equal probability to be measured in states $|00\rangle$, $|01\rangle$, $|10\rangle$, and $|11\rangle$. **(b)** 50% probability to be measured in state $|00\rangle$ and 50% for state $|10\rangle$. **(c)** 50% probability to be measured in state $|01\rangle$ and 50% for state $|11\rangle$. **(d)** 25% probability to be measured in state $|00\rangle$, 50% for state $|01\rangle$, and 25% for state $|10\rangle$.

To understand the basic difference between classical and quantum systems, let us consider an analogy that presents a problem to determine the position (A, B, C, or D equivalent to states 00, 01, 10, and 11) of an object at a given instance. Figure 6 shows an example of how the solution approaches differ for classical and quantum systems. Suppose that every state corresponds to the position of the object. In that case, the classical system determines the object's position by sequentially checking each of the possible states of the solution and validating whether the object is present. It ends up with one single state as the final solution, which in the example is position B (state 01), with 100% probability that the object is at that position (state).

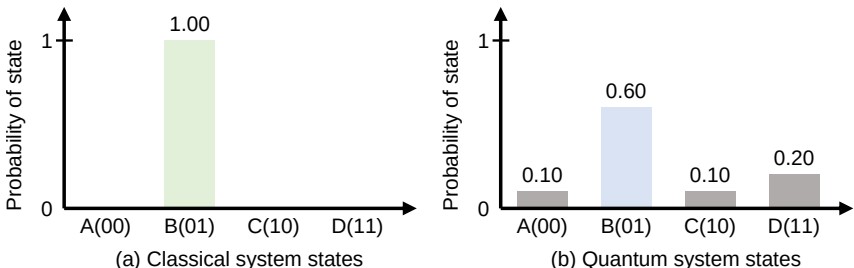

Figure 6: Example of the probability of states at one point in time for **(a)** a classical system and **(b)** a quantum system (over $n$ experiments/shots). This example helps to understand the differences between the two systems. We assume that each state shows a particular position where an object can be, i.e., positions A, B, C, or D. Both classical and quantum computers give the same solution. The main difference is that the classical system finds the solution after sequential iterations on each possible state. On the other hand, the quantum system gives a probability distribution of where the object is positioned. Both ultimately convey that the position of the object is B.

On the contrary, a quantum system solving the same problem yields a probabilistic distribution over possible measurement outcomes due to the intrinsic uncertainty of quantum states, rather than simply classical randomness. This uncertainty originates from the principles of quantum mechanics, particularly the

concept of superposition, and differs fundamentally from the statistical uncertainty described by classical thermodynamics or classical probability theory. When measured, the quantum state collapses into one of its eigenstates, such as position B (state $|01\rangle$), with a certain probability (e.g., 60%). Unlike classical uncertainty, which arises from incomplete knowledge of a deterministic state, quantum uncertainty is inherent to the quantum state itself and persists even with complete knowledge of the system's wavefunction.

### 2.3 Quantum Gates

Quantum logic gates are one of the essential parts of a quantum computer and are the building blocks of all quantum algorithms. Quantum gates are mathematically described by unitary operators that manipulate qubit states. By unitary we mean that it meets the $U^\dagger = U^{-1}$ condition, where $U$ is the gate operation matrix and $U^\dagger$ is the adjoint of $U$. This property also ensures that it is a reversible gate operation (Barenco et al., 1995). Common single-qubit gates include the Pauli-X, Pauli-Y, Pauli-Z, Rotation-X, Rotation-Y, Rotation-Z, and Hadamard gates. Multi-qubit gates, such as the Controlled-NOT (CNOT) gate, enable entanglement between qubits.

### 2.4 Quantum Circuits

Quantum circuits are sequences of quantum gates applied to qubits to perform computations. An example of a simple quantum circuit is shown in Figure 7, illustrating the application of gates and measurement operations. The actual structure of a quantum circuit, the number and the types of gates, as well as the interconnection scheme, are dictated by the unitary transformation, U, executed by the circuit (Marinescu & Marinescu, 2012b). A circuit is an independent module composed of gates arranged in a certain way across qubits to perform a task. Hence, its functionality is similar to what we refer to as an algorithm. A collection of circuits are

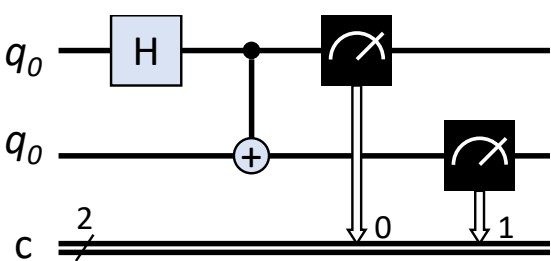

Figure 7: An example of a quantum circuit depicting input qubits, gates, measurement, and output.

independent functional quantum units using the output of a circuit as the input of another one to form a complete algorithm for the solution of the given problem (Muñoz Coreas & Thapliyal, 2022).

### 2.5 Quantum Correlations

*Correlation*, a term derived from statistics, measures how much knowledge of one part of a system can predict the behavior of another part. In classical mechanics, this correlation is deterministic: knowing the state of one part allows precise prediction of the other part's state. Any observable in a classical system has only one possible outcome, and deviations from this outcome are attributed to measurement errors or inaccuracies.

In contrast, quantum mechanics introduces a probabilistic nature to correlation. Observables in a quantum system do not have a single, definite outcome (Adesso et al., 2016). Instead, they are represented by mathematical operators whose possible measurement outcomes are described by their *eigenvalues* (Griffiths & Schroeter, 2018). Each observable corresponds to a Hermitian (self-adjoint) operator, ensuring these eigenvalues are real numbers and physically measurable (Sakurai & Napolitano, 2020). When measuring a quantum state, the system collapses probabilistically into one of the eigenstates associated with these eigenvalues. This probabilistic behavior arises not merely from randomness, as seen in classical thermodynamics, but fundamentally from the quantum principle of superposition, where a quantum state can exist as a linear combination of multiple eigenstates prior to measurement (Ballentine, 2014). Consequently, repeated measurements of identically prepared quantum systems generally yield varying results according to well-defined probability distributions inherent to their quantum state representation in a Hilbert space (Griffiths & Schroeter, 2018).

### 2.5.1 Quantum Entanglement

Quantum entanglement is a central concept in quantum information theory and represents one of the most counterintuitive aspects of quantum mechanics. Entanglement occurs when two particles become linked such that the state of one particle instantaneously influences the state of the other, regardless of the distance separating them (Bussandri & Lamberti, 2021; Cacciapuoti et al., 2020; Illiano et al., 2022). Formally, the entanglement is a quantum phenomenon where the states of two or more qubits become correlated such that the state of one qubit cannot be described independently of the others. Measurement of one qubit in an entangled pair instantaneously determines the state of the other, regardless of the distance separating them. This phenomenon, which Albert Einstein famously referred to as a "spooky action at a distance", defies the classical notion of local realism (Einstein et al., 1971).

The mechanism of quantum entanglement is illustrated in Figure 8, showing how changes in one particle affect its entangled partner. While entanglement itself is nondeterministic, meaning the specific outcomes cannot be predicted with certainty, achieving deterministic control over entangled states could lead to groundbreaking applications across various fields. Quantum development tools like Qiskit (Qiskit contributors, 2023) and PennyLane (Bergholm et al., 2018) enable researchers to create and manipulate entangled states for solving real-world problems.

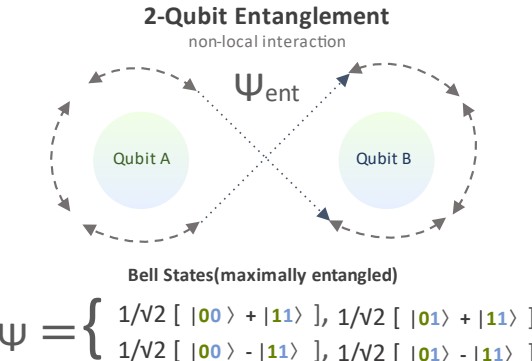

Figure 8: Overview of two-qubit entanglement. The bell states are four specific maximally entangled (shared) quantum states of two qubits.

### 2.5.2 Quantum Decoherence

A quantum system exhibits *coherence* when there is a well-defined phase relationship between its different states. Coherence is crucial for quantum computing because it determines how long a qubit can maintain its state without being disturbed by external factors. High coherence allows for extended computations, as the qubit can retain its information over the necessary duration for processing.

Quantum *decoherence* is the process by which a quantum system loses its coherence due to interactions with the external environment, leading to errors in quantum computations. This loss of coherence manifests as the system's transition from a pure quantum state to a mixed state, effectively causing a collapse of the wave function (Schlosshauer, 2019). Decoherence can be viewed as the leakage of information from the quantum system into the environment, or as environmental interference disrupting the quantum state.

Decoherence presents a significant challenge in developing scalable and stable quantum computers. It limits the duration for which qubits can reliably perform computations, thereby affecting the overall performance and viability of quantum systems. Addressing and mitigating quantum decoherence is a primary focus in the quest to build practical quantum computers.

### 2.6 Quantum Noise

Leading from the idea of coherence and decoherence, *quantum noise* encompasses various error sources (with corresponding noise channels), including bit-flip, phase-flip, amplitude damping, phase damping, and depolarizing errors. Quantum noise in realistic devices arises when qubits interact with uncontrolled environments, leading to non-unitar evolution that cannot be described by simple Schrödinger dynamics (Nielsen & Chuang, 2010b). Due to noise, the implementation of large-scale and reliable quantum computers with low error rates becomes extremely challenging. Therefore, it is imperative to properly characterize the noise in quantum systems and devise mitigation techniques to detect and correct the errors (Shaib et al., 2021).

In contrast to pure-stat descriptions, density matrices capture classical and quantum uncertainties, encoding both mixedness and coherence. Noise channels map density matrices to more mixed states (higher entropy), degrading off-diagona elements that represent quantum coherence (Nielsen & Chuang, 2010b; Lidar & Birgitta Whaley, 2003). This loss of coherence limits circuit depth and algorithmic performance on NISQ devices, making it essential to quantify noise strength via measures such as diamond norms or average gate fidelities (Magesan et al., 2011).

### 2.6.1 Noisy Intermediate-Scale Quantum Era

The long-term goal for QC is to develop functional quantum algorithms that can solve problems despite the noisy environmental systems they work in. Completely eliminating the noise errors is extremely difficult. Hence, the community has defined a set of achievable intermediate goals to evolve from Noisy Intermediate-Scale Quantum (NISQ)[1] to Fault Tolerant Quantum Computers (FTQC); see Figure 9.

The current state of quantum computing is referred to as the NISQ era (Preskill, 2018). Current quantum processors only support 50-100's of qubits but are not advanced enough to guarantee complete fault tolerance. Despite that, they represent a valid infrastructure for experimenting and improving the designs towards ideal systems. In the NISQ era, near-term hybrid quantum-classical algorithms are designed and applied to various fields, such as quantum chemistry, QML, and combinatorial optimization.

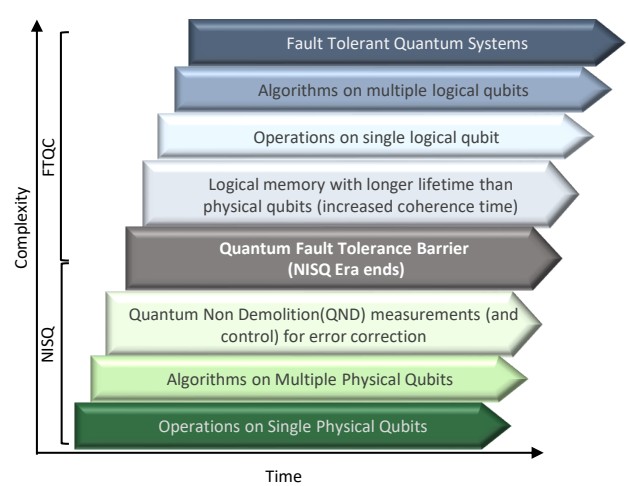

Figure 9: Quantum Information Processing Development Stages highlighting NISQ and FTQC era phases.

Contemporary quantum hardware remains limited in scale and fidelity. For example, IBM's Eagle processor contains 127 superconducting qubits, and a recent IBM chip (Osprey) achieved 433 qubits (IBM), while IonQ's latest Forte trapped-ion system operates with 32 qubits (with coherence times exceeding 1 second) (Chen et al., 2023). However, superconducting qubit devices typically have coherence times on the order of only tens of microseconds and two-qubit gate error rates around $10^{-3}$–$10^{-2}$ per gate, whereas trapped-ion qubits can maintain quantum coherence for durations on the order of seconds with single- and two-qubit gate errors on the order of $10^{-4}$–$10^{-3}$ (Chen et al., 2023). Furthermore, superconducting architectures generally allow only limited connectivity between qubits (each qubit usually interacts directly with only 2–3 neighbors in a heavy-hex lattice topology), which means that additional swap operations are required to entangle distant qubits. Each such operation adds noise and overhead. These hardware constraints, such as limited qubit count, finite coherence, gate errors, and restricted connectivity, impose strict limits on the depth and size of quantum circuits that can be executed reliably. As a consequence, quantum algorithms in the NISQ era must be shallow and noise-aware, and even then their performance and scalability are significantly curtailed by hardware noise and errors. For instance, a state-of-the-art trapped-ion system can execute on the order of 500 two-qubit gate operations in a single circuit before decoherence or errors become prohibitive, whereas many superconducting processors can only perform on the order of tens of two-qubit gates before accumulated errors overwhelm the results. Such comparisons illustrate the gap between current hardware capabilities and the requirements for practically useful quantum algorithms. . While not yet capable of fault-tolerant quantum computation, NISQ devices enable the exploration of quantum algorithms and applications in areas such as optimization, chemistry, and machine learning.

---

[1]NISQ is a term coined by John Preskill (Preskill, 2018).

### 2.6.2 Error Correction

Due to the need for algorithms that are capable of handling quantum noise in the NISQ era, several techniques have been proposed for error correction and mitigation. In classical systems, error correction mechanisms are based on redundancy. If the copies do not retain the same value, a majority vote determines the correct value. This process works well in systems with a sufficiently low error probability, since it is most likely that only single errors appear.

Similarly, error correction techniques for quantum systems do not correct 100% of the errors but help to reduce the effect of the noise. Unlike classical bits that can only be affected by a flip between 0 and 1, qubits can also experience phase errors, i.e., errors appearing when a qubit state changes its phase. Moreover, quantum errors are continuous since the rotation angle can assume any value.

Due to the no-cloning theorem (Wootters & Zurek, 1982), creating a perfect copy of a quantum state is impossible. Hence, Quantum Error Correction (QEC) codes introduce redundancy by spreading the information of a single qubit onto an entangled state of multiple physical qubits. With this approach, it is possible to perform multi-qubit (syndrome) measurements to extract the information about the error without altering the quantum information. Typical QEC schemes are the three-qubit bit flip code (Peres, 1985), the three-qubit phase flip code (Nielsen & Chuang, 2010a) and Shor's nine qubit code (Shor, 1995).

As shown in Figure 10, it is currently not possible to engineer systems with noise rates lower than $10^{-2}$ or $10^{-3}$ per gate, but the quantum community is confident that this threshold can be overcome in the next 5-10 years.

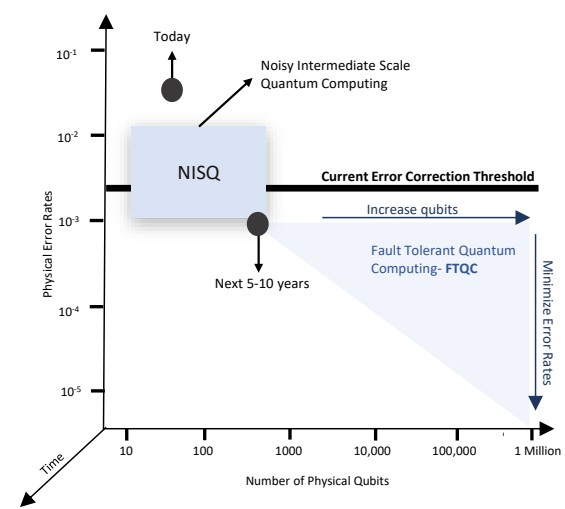

Figure 10: Current state of QEC development. The figure shows the physical error rate threshold per gate is currently not possible to overcome in the NISQ era.

### 2.6.3 Error Mitigation

Despite sharing the same goal as QEC to reduce the impact of quantum noise, Quantum Error Mitigation (QEM) techniques operate differently. While QEC aims to restore the correct value after the error occurs, QEM aims to reduce or suppress the errors that occur during computation, without full error correction. Moreover, as shown in Figure 11, QEC completely removes the noise, while QEM keeps a small amount of noise under control. A scenario where QEC is applied, called 'hard regime', is theoretically ideal but difficult to realize. The other scenario where QEM is employed,

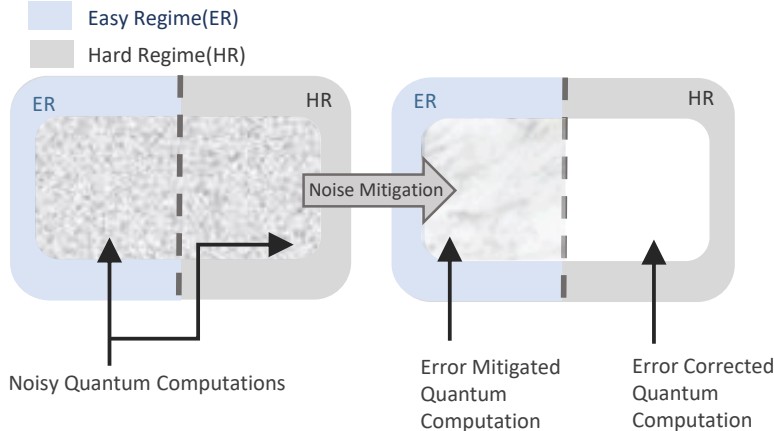

Figure 11: Error correction vs. error mitigation.

called 'easy regime', is experimentally easier to implement but at the price of lower qubit protection. For QEM deployment, Mitiq (LaRose et al., 2022) is an open-source error mitigation package in Python that

implements QEM techniques like zero-noise extrapolation (Giurgica-Tiron et al., 2020) and probabilistic error cancellation (van den Berg et al., 2023) for quantum computers.

The NISQ era is characterized by relatively small-sized quantum circuits. Moreover, quantum noise affects state preparation, gate operations, and measurement. Due to the many qubits and large circuit depth required by QEC codes, it is impossible to implement QEC on NISQ devices. On the other hand, QEM techniques offer low-overhead solutions to implement quantum circuits in an accurate and reliable way (Shaib et al., 2021; Nielsen & Chuang, 2010a).

### 2.6.4 Fault-Tolerant Quantum Computing

Fault tolerance is a property that enables a system to continue operating not only in normal conditions but also in the presence of hardware or software failures. Advancements in hardware and error correction are essential for transitioning from the NISQ era to scalable, fault-tolerant quantum computing. While there is not often a direct comparison between classical and quantum information processing, the principles of fault tolerance in classical systems are easily transferable.

In a quantum computer, the basic gates are much more vulnerable to noise than classical transistors, since qubits' implementations depend on manipulating single electron spins, photon polarization, or similar fragile subatomic particle systems. Currently, the main challenge for developing efficient and reliable quantum hardware is the ability to maintain qubit states long enough to perform useful computations without requiring redundancy efforts that cause high compute inefficiency and overhead due to the duplication of states. However, it might not be possible to engineer systems with lower error rates. An important ongoing research direction dictates investigating alternative technologies for qubit materials to build quantum systems that achieve reasonable coherence times.

Additionally, the phenomenon of entanglement makes quantum systems inherently fragile. The interactions between qubits during the execution of the quantum circuit lead to errors. If several errors appear in an uncontrolled manner, the QEC or QEM is overwhelmed, and the computation will fail. Therefore, the main goal of FTQC is to control the propagation of faults between processes.

### 2.7 Quantum Stack

To facilitate understanding quantum computing systems, it is helpful to consider them as structured in a hierarchical stack of abstraction layers, similar to classical computing architectures. As depicted in Figure 12, the quantum computing stack is generally composed of several distinct levels: quantum hardware (physical qubits and their implementation), quantum control (gate operations and measurement), quantum compilation (translation of quantum algorithms into hardware-specific gate sequences), and quantum algorithms and applications (high-level problem-solving procedures exploiting quantum properties).

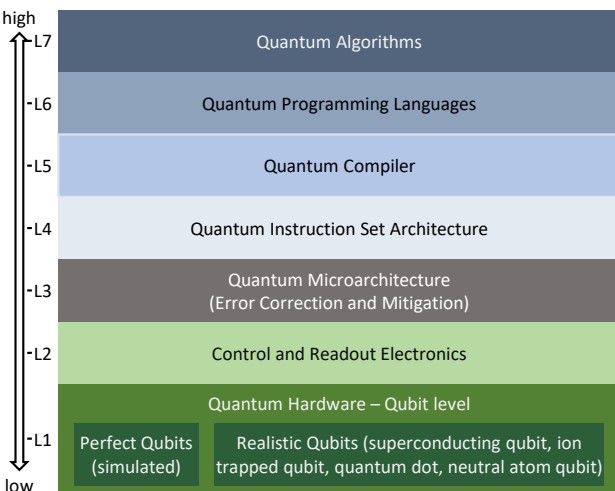

Figure 12: Quantum stack - from Hardware to Applications.

The lowest level of this stack consists of quantum hardware, which includes the physical realization of qubits using technologies such as superconducting circuits, trapped ions, quantum dots, or neutral atoms (Bertels et al., 2021).

Above this lies the quantum control level, responsible for precise manipulation and measurement of qubit states. Quantum compilation translates algorithmic descriptions into hardware-specific operations, optimizing quantum circuits to minimize resource usage and mitigate errors. Finally, at the highest level are

quantum algorithms and applications, which leverage quantum mechanical principles such as superposition and entanglement to solve complex computational problems efficiently.

This hierarchical model of the quantum computing stack provides a structured perspective for understanding how high-level quantum applications ultimately interact with underlying physical hardware, helping researchers and practitioners navigate the complexities involved in developing quantum computing systems.

# 3 Benefits of Quantum Computing

This section discusses key advantages of quantum computing compared to classical computing, emphasizing complexity-theoretic insights, quantum supremacy, and computational speedups. Understanding these benefits motivates ongoing research and development in quantum computing technologies.

## 3.1 Quantum Complexity Classes and Speedup

Quantum computing provides potential advantages over classical computing through its fundamentally different computational complexity characteristics. Classical complexity theory classifies problems into well-known classes such as P, NP, and PSPACE based on their solvability and computational resource requirements (Papadimitriou & Yannakakis, 1991) (see Appendix A for more details). Quantum computing expands this framework by introducing the complexity class *Bounded-error Quantum Polynomial-time* (BQP), consisting of problems efficiently solvable by quantum computers with bounded error probabilities (Watrous, 2009). As shown in Figure 13, the BQP class extends beyond classical P, potentially encompassing problems that are believed to be classically intractable. This perspective provides a formal basis for identifying computational problems where quantum algorithms may yield significant advantages.

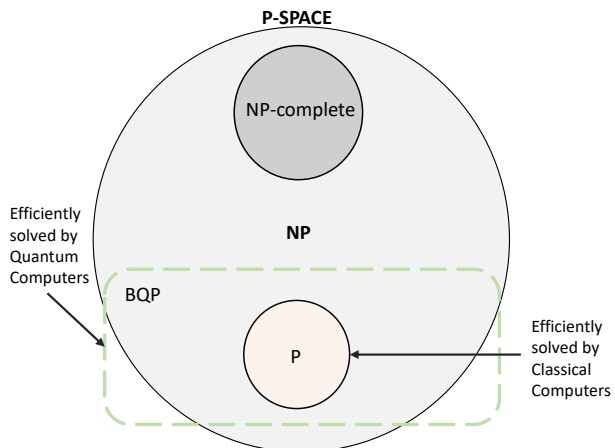

Figure 13: Quantum Computing complexity theory. The classical complexity theory is extended with the class of Bounded-error, Quantum, and Polynomial-time (BQP) problems.

## 3.2 Quantum Supremacy

"Quantum supremacy" refers to the demonstration that quantum computers can perform certain computational tasks significantly faster than any known classical algorithm (Frank Arute, 2019). Early demonstrations focused on *random circuit sampling*, generating bit-strin outputs from a randomly chosen quantum circuit drawn from an ensemble believed to be hard to simulate classically (Boixo et al., 2018). The theoretical hardness of this task is grounded in complexity-theoreti conjectures (e.g., non-collaps of the polynomial hierarchy) rather than a proof that classical algorithms cannot ultimately match performance (Aaronson & Arkhipov, 2011). For instance, Google's Sycamore quantum processor (Frank Arute, 2019) achieved a computational task in approximately 200 seconds, whereas the estimated classical computation would require about 10,000 years. Such demonstrations highlight the practical scenarios in which quantum devices exhibit exponential speedups over classical hardware, thus marking a critical milestone toward realizing practical quantum computing. However, subsequent advances in classical simulation, such as tensor network contraction optimizations and improved sampling algorithms, have dramatically reduced the estimated classical runtimes (Zhao et al., 2025). This evolution underscores that quantum supremacy benchmarks serve more as stress tests of classical simulation methods than as definitive proof of intractability.

### 3.3 Polynomial and Exponential Quantum Speedups

Quantum algorithms offer computational speedups over classical counterparts that are generally categorized as either exponential or polynomial. Exponential speedups, such as Shor's algorithm for integer factorization (Shor, 1997), demonstrate drastic performance improvements, turning classically infeasible problems into quantum tractable tasks. In contrast, polynomial speedups, exemplified by Grover's search algorithm (Grover, 1996), enhance performance significantly but do not reduce exponential complexity to polynomial. Understanding the types and implications of these speedups provides crucial insights for effectively leveraging quantum computing.

### 3.4 Practical Quantum Enhancement in Hybrid Quantum-Classical Approaches

In the NISQ era, quantum computing primarily complements classical computing rather than fully replacing it. Hybrid quantum-classical algorithms exploit quantum systems to accelerate specific sub-tasks within larger classical computations (Preskill, 2018). Recent work demonstrates quantum enhancements for practical computational problems, particularly in machine learning and optimization tasks (Pokharel & Lidar, 2023; Liu et al., 2021).

In light of the hybrid quantum-classical systems, the work of Liu et al. (2021) discusses important aspects that should be changed to obtain meaningful developments and revolves around how heuristic-based algorithms do not provide formal evidence that showcases their genuine and consistent advantage over classical algorithms. The work of Kashif et al. (2024a) demonstrates that HQNNs exhibit better scalability than classical neural networks when increasing classification task complexity. Moreover, the variational circuits can only implement linear classifiers on input quantum mechanically encoded to improve feature extraction, which can be replaced by classical support vector machines if the encoding is classically tractable. These hybrid strategies leverage quantum strengths in carefully selected areas, highlighting a realistic and immediate path toward practical quantum computing applications.

### 3.5 Quantum Information Encoding Efficiency

Quantum computers leverage superposition and entanglement to represent and process complex datasets more compactly than classical systems. Due to the superposition phenomenon, a qubit can represent the dual state of 0 and 1 simultaneously, while a classic binary bit can only take either 0 or 1 at a time. Hence, to express n-bit combinations using a classical computer, we need $2^n$ combinations, which can be checked sequentially before the classical computer can find the solution. On the other hand, a quantum computer takes advantage of its wave nature so that the wave interference increases the probability of the desired state and decreases the other to reach a correct solution effectively all at once.

### 3.6 Large-Scale Infrastructures and Design Automation

Infrastructure is an asset to any economy. Its development and maintenance, despite being difficult, have an enormous contribution to economic development and prosperous future growth. Continuous needs for humanitarian survival and improved quality of life are attracting necessary investments in transportation, civil engineering, and health sectors. Developing systems that provide smooth processing for these domains is challenging, given the burden of data required to process and handle with the aspects of time criticality, user safety, and reliability. The potential of QC to provide extensive grounds to optimize these processes defines new out-of-the-box ways for infrastructural development that are entirely new and creatively solve the problems at hand in a way that was impossible with classical computers.

The realization of large-scale quantum systems will take several years. However, it is the ultimate target for all significant quantum hardware companies. Their roadmaps aim to have such large-scale systems available for industrial and practical use around the end of the NISQ era. Figure 14 shows the development roadmaps provided by IBM (Quantum, 2022) and Honeywell (Honeywell, 2022a) for their current progress and plans towards achieving large-scale quantum infrastructures. These milestones demonstrate the continuous focus on developing quantum chips with more qubits every year, like the 432-qubit Osprey machine by IBM (IBM,

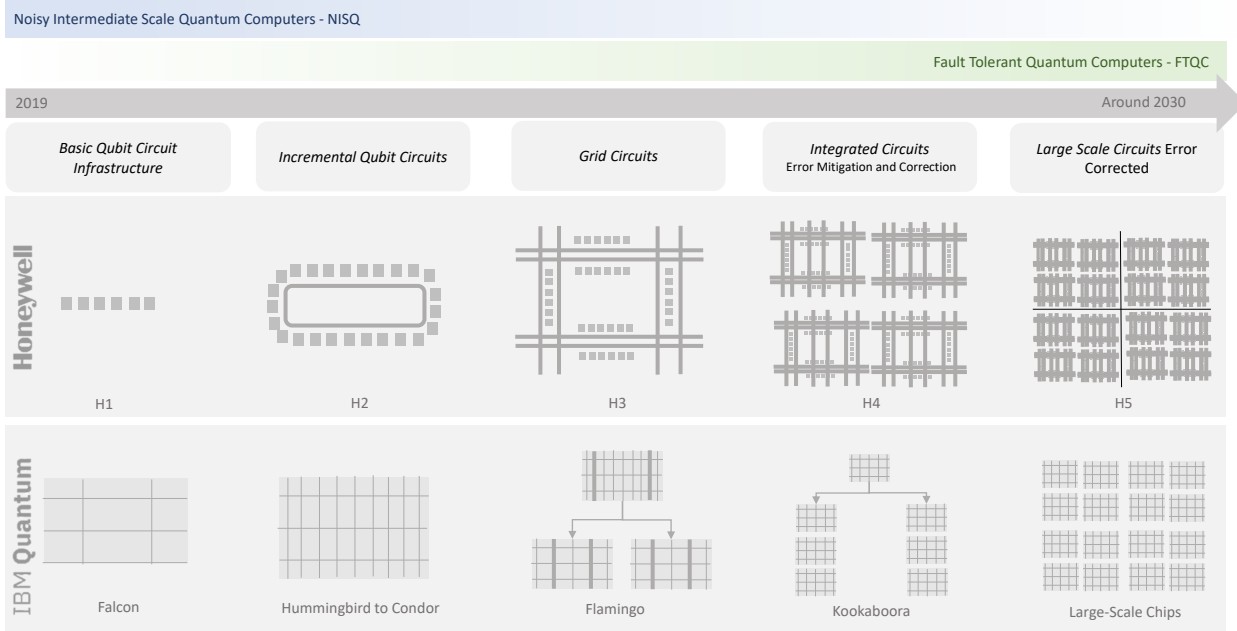

Figure 14: Roadmap to design large-scale quantum chips by Honeywell and IBM. Starting from the basic qubit circuit infrastructure, the systems incrementally evolve to support more qubits. The development roadmap includes grid circuits and integrated circuits with error correction and mitigation, to ultimately design large-scale circuits that are envisioned to be released by 2030.

2022). Similarly, Honeywell's H-series (Honeywell, 2022b) has stages planned from H1 to H5 as their progress from 10 qubit systems to larger scale qubit quantum systems envisioned to be designed around the year 2030.

## 4 Quantum Machine Learning Algorithms

This section provides an overview of QML algorithms and their domain applicability. ML is a class of advanced algorithms that perform a certain task. Given a large number of inputs and desired outputs, an ML model can be trained to make predictions on unseen data. If it is executed on quantum computers, it becomes a quantum ML algorithm. An overview of the existing QML algorithms is shown in Figure 15.

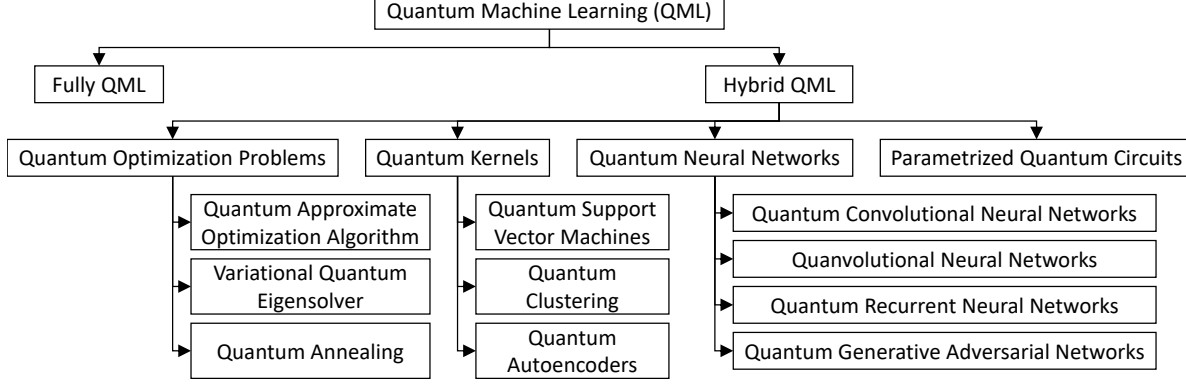

Figure 15: Overview of QML algorithms.

### 4.1 Categorization of QML Approaches

Before diving into the details of QML algorithms, it is important to characterize different approaches based on the type of data and type of processor used to solve the problem (Aïmeur et al., 2006). The four categories (see Figure 16) are formed based on whether the data is classical (C) or quantum (Q) and whether the algorithm runs on a classical (C) or quantum (Q) computer.

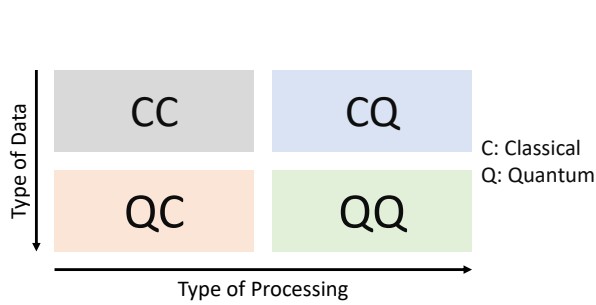

Figure 16: Categories of QML based on types of data and processor.

Figure 17: Example of a parametrized quantum circuit.

- **CC** refers to processing *Classical data using Classical computers*, but using algorithms inspired by quantum computing, such as the recommendation system algorithm.

- **CQ** refers to processing *Classical data using Quantum machine learning algorithms* and will be the main focus of this chapter.

- **QC** refers to processing *Quantum data using Classical machine learning algorithms*. This is an active area of investigation, with classical machine learning algorithms used in many quantum computing areas, such as qubit characterization, control, and readout

- **QQ** refers to processing *Quantum data using Quantum machine learning algorithms*. It is also known as Fully Quantum Machine Learning (FQML). It can be considered a future investigation area that can be developed during a more mature stage of quantum computing.

### 4.2 Parametrized Quantum Circuits

Variational or Parametrized Quantum Circuits (PQCs) are specific types of quantum algorithms that depend on free parameters. PQCs allow us to utilize the existing quantum computers to their full extent. In the context of QML, PQCs are used either to encode the data, where the parameters are determined by the data being encoded, or as a quantum model, where the parameters are determined by an optimization process. PQCs can be interpreted as ML models, considering a variational quantum classifier that uses two variational circuits (see Figure 17). The first circuit associates the gate parameters with fixed data inputs, while the second circuit depends on free and trainable parameters. This setup provides the basic building blocks to build QML algorithms on NISQ devices (Benedetti et al., 2019).

A Variational Quantum Algorithm (VQA) combines a classical optimizer with a PQC. VQAs represent a promising approach to achieving quantum advantage over classical systems, even when using NISQ devices that come with various constraints. These constraints include a limited number of qubits, restricted qubit connectivity, and decoherence errors that limit the depth of quantum circuits (Cerezo et al., 2021a). Despite these challenges, VQAs can leverage the strengths of both classical and quantum computing to tackle problems that are intractable for classical algorithms alone. Examples of applications in which VCAs excel are quantum chemistry, material science, financial modeling, and cryptography.

### 4.3 Quantum Optimization Problems

An optimization problem aims to find the best solution to a given challenge. For instance, the goals of a business are to minimize the production cost or maximize its revenue. The solution can either be discrete if the variable to optimize is determined from a countable set or continuous if the optimized value is found from a continuous function. It can also be either single-objective or multi-objective. Applying QML to solve optimization problems enables us to employ quantum classification and regression programs on QML infrastructure (see Figure 18). The intrinsic parallelism of quantum computing can speed up the optimization computation to compute the global minimum (or maximum) faster than classical computing (Dasari et al., 2019).

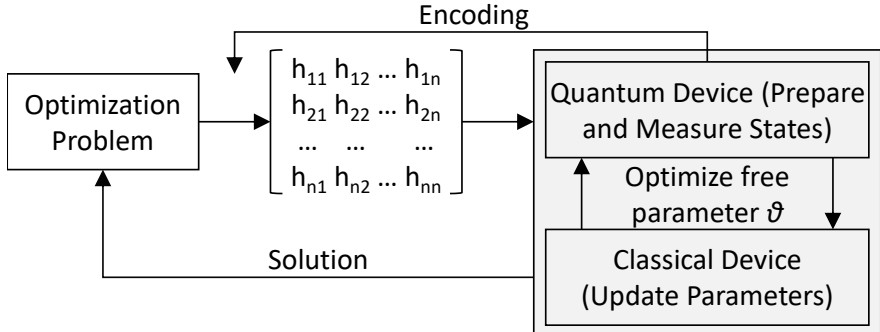

Figure 18: Solving a combinatorial Optimization Problem with hybrid quantum models. Hamiltonians (H) are always square matrices with lengths equal to $2^n$, where $n$ is the number of qubits in the Hamiltonian.

The primary goal and advantage of all efforts behind quantum algorithms is to motivate and identify the practical advantage over classical in practical scenarios. Similarly, in gate-based quantum computers, which are currently adopted in the industry, and in shared future directions, it is important and useful to define beneficial use-cases of a quantum algorithm, to showcase its efficient utilization of the quantum hardware. Such efforts are now making their way into the research work involving techniques like Filtering Variational Quantum Eigensolvers (Amaro et al., 2022). Driven by the idea of filtering operators while processing to increase speed and reliability towards converging towards an optimal solution, such efforts are necessary to make QML systems impactful.

#### 4.3.1 Amplitude Amplification and Estimation

Amplitude amplification generalizes Grover's search algorithm to boost the probability of measuring desired outcomes. This technique provides a quadratic speedup over classical brute-force search methods (Brassard & Hoyer; Grover, 1996).

Quantum Amplitude Estimation (QAE) builds upon amplitude amplification to estimate the probability amplitude of a target quantum state. The original version relies on phase estimation, which is resource-intensive, while iterative variants avoid phase estimation at the cost of more measurements (Brassard & Hoyer; Grinko et al., 2021). These methods are suited for estimating expectation values and integrals in quantum finance and chemistry.

#### 4.3.2 Quantum Approximate Optimization Algorithm

The Quantum Approximate Optimization Algorithm (QAOA) (Farhi et al., 2014) is a technique that finds approximate solutions to combinatorial optimization problems. It is based on PQCs that approximate the adiabatic evolution from an initial Hamiltonian. A set of parameters are tweaked to optimize a cost function out of the quantum circuit output. The QAOA method is a promising candidate for achieving quantum advantage over classical systems in the NISQ era. Potential applications of the QAOA in the real world span from the logistics field, such as designing air/ground traffic and shipping routes, to the finance domain, where it can be used to maximize profits and minimize risks for a given portfolio.

### 4.3.3 Variational Quantum Eigensolver

A Variational Quantum Eigensolver (VQE) is a hybrid algorithm that uses both classical computers and quantum computers to find the ground state of a given physical system. The VQE algorithm is versatile and can be applied to a wide range of tasks, making it a promising candidate for use in many applications of quantum computers, including machine learning and control theory (Tilly et al., 2022). One particularly useful application of the VQE algorithm is finding the ground state energy (i.e., the minimum eigenvalue) of a Hamiltonian. This is accomplished by minimizing a cost function based on the Hamiltonian's expectation value. An additional term in this algorithm accounts for the overlap between the excited and ground states.

An instance of VQE requires the definition of two algorithmic sub-components, which are a quantum trial state (also called *Ansatz* (Zhang, 2012)) and a *classical optimizer*. Common ansatz choices in VQE include hardware-efficient circuits and chemically inspired ones. A hardware-efficient ansatz uses layers of native quantum gates arranged in a repetitive pattern, enabling expressive states with relatively shallow circuits suited for NISQ devices (Kandala et al., 2017). In contrast, the Unitary Coupled Cluster (UCC) ansatz, popular in quantum chemistry, constructs the trial state by applying a series of particle excitation operators, producing a physically motivated but often deeper circuit (Romero et al., 2018). The optimizer varies the parameters of the Anzatz, such that it works towards a state, as determined by its parameters, that results in the minimum expectation value being measured by the input operator (Hamiltonian). Additionally, the VQE can be used to model complex wave functions in polynomial time, making it one of the most promising NISQ applications for quantum computing. In practical applications, the VQE is used in chemistry for simulations of molecules, as well as logistics and network design.

However, the performance of VQE in real-world settings is strongly influenced by the choice of ansatz and the optimization strategy. Deep or highly unstructured ansatze can lead to *barren plateaus*, i.e., vast flat regions in the parameter landscape where gradients vanish, hindering the training process (McClean et al., 2018; Cerezo et al., 2021b). The selection of a classical optimizer (e.g., gradient-based methods versus gradient-free heuristics) also impacts convergence. Certain optimizers may better navigate noise and avoid local minima, but no single optimizer works best for all problems (Tilly et al., 2022). Consequently, these factors have the following practical implications. An overly simplistic ansatz might not be expressive enough to reach the true ground state. On the other hand, an overly complex ansatz can require prohibitively deep circuits or become untrainable on noisy hardware, limiting VQE to relatively small systems in current experiments (Kandala et al., 2017; Tilly et al., 2022; Liu et al., 2024).

### 4.3.4 Quantum Annealing

The Quantum Annealing (QA) method is an algorithm to solve combinatorial optimization problems. Instead of using the temperature to explore the problem space, the QA uses the laws of quantum mechanics to measure the energy state. Starting from the qubits' dual state, where all the possible combinations of the values are equally likely to happen, the quantum mechanical effects are gradually reduced through a process called quantum fluctuation (Park & Nha, 2023) to reach the most

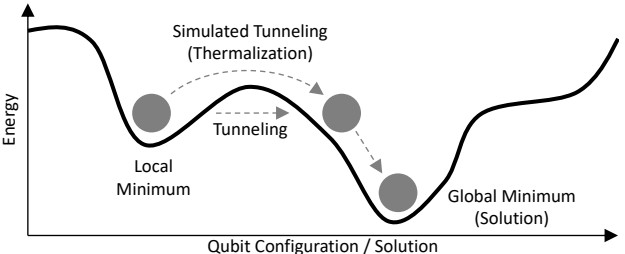

Figure 19: Example of quantum annealing.

optimal point where the lowest energy state is achieved (see Figure 19). In practical use-cases, the QA is efficient for combinatorial optimization problems whose search space has several local minima, such as the traveling salesman problem.

QA techniques require an objective function to be defined in the *Ising model* or *Quadratic Unconstrained Binary Optimisation (QUBO)*. The Ising model is a mathematical model in statistical mechanics (Borle & Lomonaco, 2019). The QUBO (Jun, 2022) is mathematically equivalent to the Ising model and can be used to formulate a problem in a more simple way.

In physics, the minimum energy principle (Callen, 1985) states that the internal energy decreases and approaches the minimum values. If we can formulate our problems as an energy minimization problem,

the QA method can search for the best possible solution by utilizing quantum methods over an energy landscape.

### 4.3.5 Key Distinctions

Unlike VQE, which uses gate-based quantum circuits and classical optimizers, QA uses adiabatic evolution and is often implemented on specialized hardware like D-Wave systems. While VQE targets general Hamiltonians using variational circuits and classical feedback loops, QA is a continuous-time analog process that exploits quantum tunneling and annealing to reach the solution. QA does not require gate-level programming, but lacks the circuit expressiveness and flexibility of VQE.

Both methods aim to approximate low-energy solutions, but differ in control mechanisms (discrete variational updates vs. adiabatic evolution) and hardware requirements. VQE is more tunable and programmable, while QA provides natural hardware-level implementation for combinatorial problems.

### 4.4 Quantum Neural Networks

Quantum Neural Networks (QNNs) are computational Artificial Neural Network (ANN) models that are based on the principles of quantum mechanics (Panella & Martinelli, 2011). As shown in Figure 20, typically in QNNs, the input data is loaded with classical data inputs. The quantum circuit contains a feature map module with input parameters and an Ansatz module with trainable weights. Measurements are conducted to obtain the outputs.

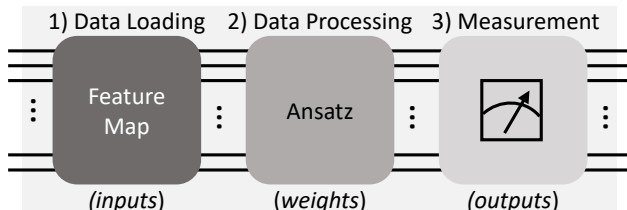

Figure 20: Example of a Quantum Neural Network.

What makes QNNs exciting compared to traditional ANNs is the differentiable nature of the quantum circuits. Quantum computers can compute the changes in the control parameters needed to make the QNN better at a given task. While common applications of QNNs are in the area of image recognition and object detection since they excel in detecting specific features from images, QNNs can be used for a plethora of applications, as discussed in Section 7.

During the NISQ era, the main focus is on Hybrid Quantum Neural Networks (HQNNs). The generic structure of an HQNN is illustrated in Figure 21, while specialized model types are discussed in the following sections.

### 4.4.1 Quantum Convolutional Neural Networks

Convolutional Neural Networks(CNNs) are established architectures for image classification tasks in the computer vision domain (Capra et al., 2020). Quantum Convolutional Neural Networks (QCNNs) are HQNN architectures whose structure is inspired by classical CNNs (see Figure 22). The QCNN circuit model proposed by Cong et al. (Cong et al., 2019) extends fundamental properties of CNNs to the quantum domain by using only $O(log(N))$ variational parameters for input sizes of $N$ qubits. Therefore, it enables efficient training and implementation of QCNNs on realistic, near-term quantum devices.

The structure of a classical CNN consists of applying alternating convolutional layers (with an activation function) and pooling layers, typically followed by fully-connected layers before the output is generated. In QCNNs, the convolution operations are performed as parameterized unitary rotations based on PQCs, executed on neighboring pairs of qubits. These convolutions are followed by pooling layers. The scope of pooling layers is to down-sample the results of the previous layer by reducing the data dimensions while inherently extracting the most relevant features. The dimensionality reduction with pooling layers reduces the dimensions of the quantum circuits, which in turn reduces the number of qubits in the circuit while retaining the maximum information possible from predecessor layers. Consequently, it reduces the computational cost of the complete circuit since the number of learnable parameters of the QCNN is equal to the qubits in the circuit. However, in quantum physics, it is not possible to

directly remove qubits from the circuit. Hence, the pooling layers in quantum circuits are deployed by measuring a subset of the qubits and using these measurement results to control subsequent operations. The fully-connected layers are implemented as a multi-qubit operation on the remaining qubits before the final measurement. All the parameters involved in these operations are learned during training.

While the classification task is the primary purpose for using QCNNs, they can also devise a quantum error correction scheme optimized for a given error model (Cong et al., 2019). However, QCNNs with large circuit depths are challenging to implement due to the high coherence required for qubits.

### 4.4.2 Quanvolutional Neural Networks

Motivated by the idea of a convolution layer where, instead of processing the entire input data with a global function, a local convolution is applied, Quanvolutional Neural Networks (QuanCNNs) (Henderson et al., 2020) are based on a "quanvolutional" kernel. Compared to QCNNs, where the complete architecture is implemented with PQCs, QuanCNNs focus only on building efficient convolutional layers using quantum circuits (see Figure 23). Considering that the underlying basic principle of convolutional layers is to extract features from data hierarchically, a quanvolutional layer leverages the ability of quantum systems to contextualize greater information capacity. Therefore, QuanCNNs introduce preliminary layers before the classical CNN to manipulate (i.e., pre-process through transformations) the inputs using a single or a set of consecutive quantum layers. The input of the first classical

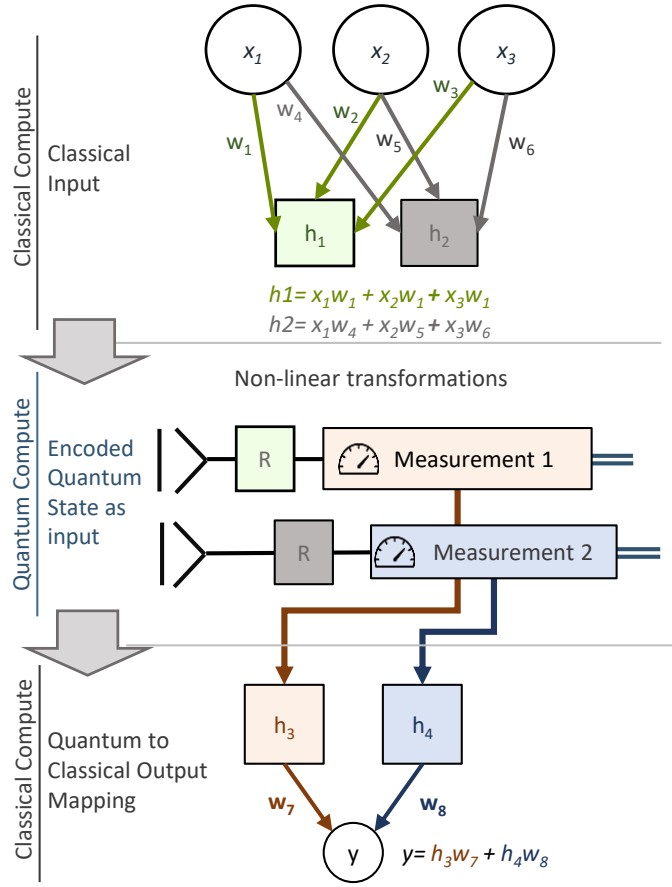

Figure 21: Functionality of a Hybrid Quantum Deep Neural Network. The classical input is converted into the quantum domain by encoding the data as a quantum state. After measurements, a quantum to classical output mapping is required to obtain the final results.

layer is a quantum output measured classically that preserves a large amount of information for improving the contextual feature extraction in the classical layers. In other words, it aims to encapsulate the input information using a representation that results from the quantum space transformations. This approach complements and facilitates the architecture in identifying correlated features in the subsequent (classical) layers.

**QCNN vs. QuanCNNs:** QCNNs implement an end-to-end deep network with a set of quantum circuits, whereas QuanCNNs use quantum circuits only for the convolutional feature extraction stage before feeding into classical layers. QCNNs leverage quantum pooling (via qubit measurement and feed-forward control) to progressively perform spatial down-sampling. However, a QCNN requires high qubit coherence for multiple sequential quantum layers, making large-depth QCNNs challenging on NISQ hardware. In contrast, QuanCNNs trade off full quantum depth for practicality. They insert one or a few quantum convolution layers as a preprocessing step to a classical CNN, exploiting quantum feature maps to encode richer representations while relying on classical layers for complex feature hierarchy. This hybrid design means QuanCNNs can work with shallow quantum circuits (low qubit counts and depths) that are feasible on current devices. In summary, QCNNs offer a more pure quantum architecture (potentially yielding quantum speedups or non-classical feature extraction) but face implementation limits due to noise and decoherence, whereas

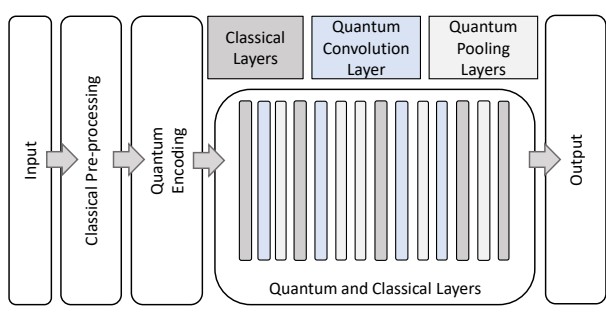

Figure 22: Functionality of a Quantum Convolutional Neural Network.

Figure 23: Functionality of a Quanvolutional Neural Network.

Quanvolutional CNNs are hybrid, easier to realize on near-term machines, and still enhance classical models by injecting quantum-computed features (often improving accuracy on image data with minimal quantum resources) (Zaman et al., 2024a).

### 4.4.3 Qauntum Recurrent Neural Networks

Recurrent Neural Networks (RNNs) are the foundational models for using sequential data to solve temporal problems such as natural language processing. Inspired by the structure of a VQE quantum circuit, a Quantum Recurrent Neural Network (QRNN) replaces the RNN layers with PQCs. While VQE circuits are very dense, QRNN cells are highly structured circuits with fewer parameters that are reused. Each parameter has a higher-level logical unit than the VQE components (Bausch, 2020). Applying multiple QRNN cells iteratively on the input sequence generates a QRNN (see Figure 24) that behaves similarly to classical RNNs. However, due to the decoherence issues, the QRNNs' performance on long sequential data is typically worse than classical RNNs. The coherence constraints can be relaxed if the QRNN architecture is built by stacking the recurrent blocks in a staggered way (Li et al., 2023).

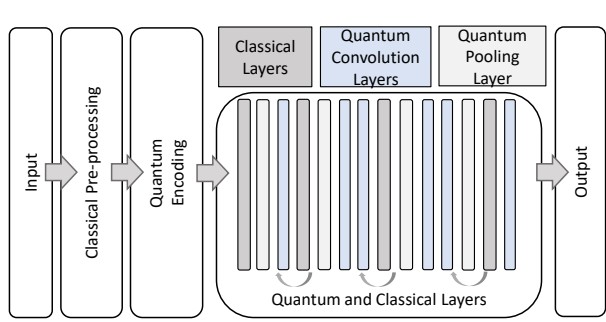

Figure 24: Functionality of a Quantum Recurrent Neural Network.

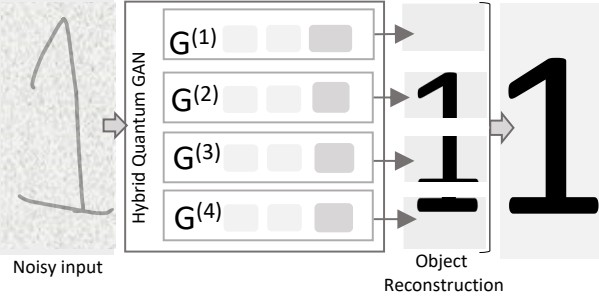

Figure 25: Functionality of a Quantum Generative Adversarial Network, where the quantum generator is split into multiple sub-generators that craft patches of the full image.

### 4.4.4 Quantum Generative Adversarial Networks

A Generative Adversarial Network (GAN) aims to generate data that resembles the original samples used in the training dataset. This is typically implemented using a generator that produces new (fake) data and a discriminator that distinguishes fake from real data. Fully Quantum GANs (QGANs) typically require too large resources that cannot be implemented in near-term quantum devices. To overcome this limitation, it is possible to design hybrid QGANs where either the generator or the discriminator is implemented with classical computing. For example, the QGAN architecture proposed by Huang et al. (Huang et al., 2020) has a quantum generator and a classical discriminator. The generator is divided into multiple sub-generator

blocks, each of them responsible for generating a patch of the full image. As shown in Figure 25, the generated image is formed by concatenating the patches. This approach can be implemented on quantum computers with a limited number of available qubits, where the same device can execute the sub-generator sequentially. This architecture provides guidance for developing advanced QGAN models on near-term quantum devices and opens up an avenue for exploring quantum advantages in various GAN-related tasks.

### 4.4.5 Quantum Autoencoders

Autoencoders (AEs) are self-supervised ML models that reduce the size of the input data by reconstructing it. It can compress and encode information from the input using representation learning. Quantum counterparts of AEs, called Quantum Autoencoders (QAEs) (Romero et al., 2017), aim to reduce the dimensionality of quantum states. As shown in Figure 26, a QAE is composed of three layers, which are input, bottleneck, and output. Its primary use is for digital compression, where the information can be encoded into a smaller amount of qubits. QAEs enable the transformation and mapping of large problems into smaller quantum circuit equivalents. Moreover, QAEs are useful for denoising since they extract relevant features from the initial quantum state into encoded data while neglecting the additional noise (Shaib et al., 2021).

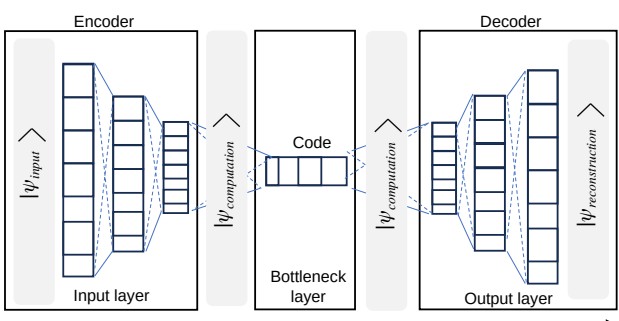

Figure 26: Functionality of a Quantum Autoencoder, where the input state is compressed through the encoder and then decompressed through the decoder.

### 4.4.6 Binary Neural Networks with Quantum Optimization

Another class of neural networks, whose design is motivated by the need for a compact architecture for increased portability and ubiquitous computing, is the Binary Neural Networks (BNNs). As suggested by their name, the working principle is to replace floating point feature weights and activations with binary values, introducing a binarization process. The redesigned strategy is favorable as it reduces the required storage and computation for resource-constrained devices while leveraging the trade-off with its performance. This process allows compact, lightweight, and power-efficient networks that can simultaneously maintain acceptable accuracy. Methods for improving the performance of BNNs are an active area of research and quantum methods theoretically present themselves as a useful tool for improvement. The work presented in Liao et al. (2019) argued theoretically that QCs could train BNNs to their global optimum via quantum search techniques in a shorter time than using brute-force classical optimization. An alternative approach to training BNNs utilizes a quantum superposition of weights (Alarcon et al., 2022; Carrasquilla et al., 2023).

### 4.4.7 Single-Qubit Neural Networks

Single-qubit QNNs are based on the data re-uploading technique (Pérez-Salinas et al., 2020). According to this method, each layer receives a copy of the input on the same single qubit line. The measured state of the solution stores the sequentially-encoded processing result carried on from all the sequential layers. The work in Yu et al. (2022) presents a theoretical framework formulated to showcase the expressive capability of the employed data re-uploading technique over the quantum neural network layers, each of them comprising interconnected encoding circuits and trainable circuit blocks. It is further highlighted how single-qubit QNNs can approximate any univariate function by mapping the model to a partial Fourier series. While it seems to be a useful direction to be explored, as demonstrated in Peña Tapia et al. (2023), there are limitations for evaluating multivariate functions by analyzing the frequency spectrum and the flexibility of Fourier coefficients which must be kept in mind while designing architectures based on the single qubit QNN template (Yu et al., 2022; Goto et al., 2021; Pérez-Salinas et al., 2021).

## 4.5 Quantum Federated Neural Networks

The base principle of Federated Learning is to conduct local training and update the global model through global aggregation. In this way, the local users have only access to a subset of the data, to maintain data privacy and high accuracy. Similarly, Quantum Federated Neural Networks (QFNNs) (Innan et al., 2024a) use QNNs in an FL environment, demonstrating high accuracy for various applications.

Figure 27: Functionality of a Quantum Federated Neural Network framework.

## 4.6 Quantum Kernels

Kernel methods are a collection of pattern analysis algorithms that employ kernel functions to operate in a high-dimensional feature space, such as Support Vector Machines (SVMs) and clustering techniques. The main objective of SVMs is to find decision boundaries to group a given set of data points into classes. When these classes' data spaces are not linearly separable, SVMs can benefit from using kernels to find these boundaries.

Quantum kernel methods extend this idea by leveraging quantum feature maps to encode data into a quantum Hilbert space (Havlíček et al., 2019). In a quantum kernel approach, each data point $\mathbf{x}$ is mapped to a normalized quantum state $|\phi(\mathbf{x})\rangle$ via a parameterized quantum circuit (feature map), and the kernel is defined as the overlap between two such states: $K(\mathbf{x}, \mathbf{x}') = |\langle\phi(\mathbf{x})|\phi(\mathbf{x}')\rangle|$. By using quantum states as feature vectors, it is in principle possible to access extremely high-dimensional feature spaces and intricate entanglement correlations that have no efficient classical representation. A crucial consideration is to choose feature maps that are *classically hard to simulate*, as this is believed to be necessary for achieving quantum advantage in learning tasks (Havlíček et al., 2019). If the quantum kernel can be efficiently computed or approximated by a classical algorithm, then a classical model could replicate its results, negating any quantum advantage (Schuld & Killoran, 2019). Conversely, feature maps based on classically intractable quantum circuits (for example, instantaneous quantum polynomial (IQP) circuits) can yield kernel functions that separate data in ways unattainable by known classical kernels (Havlíček et al., 2019).

Designing these quantum feature maps comes with significant challenges. On one hand, highly expressive or deep quantum embeddings may suffer from phenomena analogous to barren plateaus, such as an exponential concentration of kernel values, where the inner products between states become almost uniformly small regardless of input, effectively making the kernel uninformative for learning (Schuld & Killoran, 2019). On the other hand, circuits complex enough to be classically non-simulable tend to be more sensitive to noise and harder to implement on near-term hardware. In practice, there is a trade-off between expressivity and practicality. The feature map must be complex enough to exploit uniquely quantum features of the data, yet structured enough to avoid excessive noise and to permit effective training. Ongoing research is exploring how to optimize quantum kernels under these constraints, including techniques to tailor or learn the feature map for a given problem (Torabian & Krems, 2023). This careful balance in quantum kernel design is crucial for their performance in real-world applications, where both quantum hardware limitations and the risk of classical simulability must be managed.

Quantum clustering is a class of data clustering techniques that use the mathematical and conceptual tools of quantum mechanics. It belongs to a family of density-based clustering algorithms, where clusters are defined by areas of higher-density data points. Its design involves a scale-space probability function viewed at the lowest eigenstate of a Schrödinger equation, followed by simple analytic operations to derive a potential function whose minima determine cluster centers (see Figure 28). The design proposed by Horn & Gottlieb (2002) was applied to a two-dimensional system but is also scalable to higher dimensions. As demonstrated by Aïmeur et al. (2007), quantum clustering for different algorithms (divisive clustering, k-medians, and construction of a neighborhood graph) can obtain significant speedup compared to classical clustering.

## 4.7 Quantum Computing for Bayesian Machine Learning

Bayesian ML is based on utilizing Bayesian methods, like Gaussian processes, to carry out ML tasks. Providing information on the uncertainty of predictions is one of its greatest strengths (Zhao et al., 2019). Looking into channeling this aspect of advantage to creatively encompass the uncertainty in a quantum methodology is a promising research direction. Interesting results have been obtained connecting deep feed-forward neural networks with Gaussian processes, which eliminates the need for back-propagation while training (Zhao et al., 2019). As back-propagation itself represents one of the challenges in QML, there is great potential for the

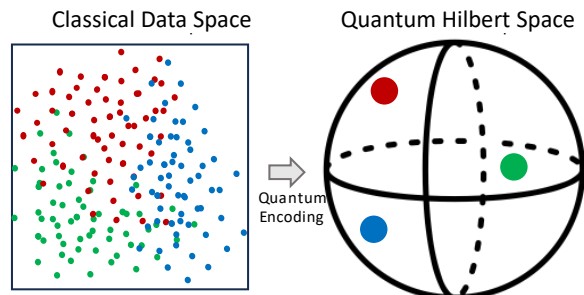

Figure 28: Overview of Quantum Clustering, where the cluster centers are represented in the Hilbert space.

success of methods such as Bayesian ML that can perform ML without back-propagation. This can be further optimized by utilizing quantum components in other useful paradigms within the framework. Specialized algorithms can find their way to implementation based on the Bayesian ML technique inspired by quantum optimizations. Such algorithms effectively exploit the connection between ML and Gaussian processes from the quantum lens to curate quantum-enhanced frameworks (Zhao et al., 2019).

## 4.8 Limitations of Current QML Approaches

Despite the recent advancements in QML, there are still some limitations due to their relatively immature technology and relatively fewer studies compared to classical ML. The following list provides the most well-known limitations of QML.

- Most of the existing QML algorithms are not fully quantum but a hybrid combination of classical and quantum operations.

- The accuracy measured by QML algorithms is not yet mature enough to be on par with the state-of-the-art classical ML algorithms.

- The lack of quantum technology and tool standardization hinders the QML algorithm development due to their different design practices.

- The high computational cost of developing large-scale QML models incurs a high demand for specialized tools and expensive computational resources.

- The performance of QML algorithms is fundamentally limited by current hardware capabilities (e.g., short qubit coherence times, limited qubit counts, and non-negligible error rates). Present processors offer only tens to a few hundred noisy qubits, with short coherence times and gate error rates often around $10^{-3}$ per operation, and many architectures have restricted qubit connectivity. These limitations severely cap the size and depth of quantum circuits that can be executed reliably. Significant advances in quantum hardware, such as longer coherence, lower error rates (via error mitigation or correction), and scaling up the number of qubits, are required for QML techniques to achieve their full potential.

- The susceptibility to noise of quantum systems dictates instability and reliability concerns.

- The barren plateau phenomenon, where gradients vanish exponentially with system size, significantly hinders the training of deeper or wider QML models such as VQEs, QCNNs, and quantum kernel methods (McClean et al., 2018). Mitigation strategies include problem-inspired ansatz design, local cost functions (Cerezo et al., 2021b), layer-wise training (Grant et al., 2019), entanglement management (Patti et al., 2021), residual connections (Kashif & Al-Kuwari, 2024), and adaptive initialization (Kashif et al., 2024b; Zhuang & Guan, 2025).

- Since the parameter update of QML algorithms relies on classical optimization modules, the quantum technology cannot be utilized to its full extent, because QML training is still bottlenecked by classical methods.

- The design strategies and usage of quantum gates in QML circuits are relatively naive and not well established.

- Realizing a quantum advantage also demands that the data encoding (feature map) and ansatz used in QML are classically hard to simulate. If a QML model's quantum states can be efficiently emulated by a classical algorithm, then the model cannot outperform classical methods (Thanasilp et al., 2024). In practice, variational circuits and quantum kernel methods must leverage feature maps that produce complex quantum states or entanglements which are infeasible to reproduce classically, as this is crucial for any genuine advantage over classical ML.

It is therefore evident that current QML techniques, such as VQEs, QNNs, and quantum kernel methods, will heavily rely on future improvements in quantum hardware. These approaches struggle with the shallow circuits and noise levels of today's NISQ devices. Hence, better qubit coherence, higher qubit counts, and lower error rates will be essential to scale up QML algorithms. Ultimately, the viability of these algorithms in solving practical problems hinges on the advent of more robust quantum processors (potentially fault-tolerant devices) that can support deeper circuits and more qubits without being overwhelmed by errors. Concurrently, continued research is needed to design quantum data embeddings and model architectures that exploit quantum complexity beyond classical simulability, ensuring that QML models can capture patterns that classical ML cannot.

## 5 Datasets

In QML, we distinguish between (i) *classical data* that must be pre-processed and encoded into quantum states for use in quantum circuits, and (ii) *quantum-native data* originating from simulations or experiments, which can be ingested more directly by QML models. Below, we separate these concerns into two parts: classical data handling (preprocessing and encoding) and quantum datasets (collections of quantum-native examples and their typical usage).

### 5.1 Data Pre-Processing

Individual independent variables operating as inputs to the ML model are referred to as *features*. They can be thought of as representations or attributes that describe the data. An ML model makes accurate and precise predictions if the algorithm can easily interpret the data's features.

However, applying ML algorithms to noisy data would not give quality results as the system would fail to identify features effectively. Noisy data would introduce factors and patterns in the learned model that are different from the actual distribution of the given problem. Therefore, data pre-processing is important to improve the overall data quality to feed the ML models. The generic data pre-processing pipeline in an ML system includes the steps needed to transform or encode data so that it can be easily parsed by the machine, as shown in Figure 29a. The same principle applies in the quantum domain, but the quantum machines should also interpret the classical data. In this regard, a coherent and structured pre-processing mechanism for QML in the NISQ era is an inevitable need. Despite the generic pre-processing pipeline applicable in any case, Figure 29b illustrates a hybrid pre-processing pipeline for quantum model development. It ensures that the information extraction is maximized using quantum machines and allows quantum computation to apply to any kind of classical data.

Since QML has not yet extensively demonstrated its advantages compared to classical ML, it is more susceptible to adoption by the industry. The current quantum computers have few qubits to test and are noisy, making it difficult to demonstrate the current and potential quantum advantage of QML methods. Utilizing quantum methods for ML in the pre-processing phase, it is possible to achieve better classical encoding and performance of quantum classifiers (Mancilla & Pere, 2022).

Since many proposed QML applications rely on using well-known datasets, especially in the context of quantum data with constraints generated due to the underlying quantum mechanics for representing it as quantum states, it is important to formulate and standardize its collection and encoding processes over time to maintain quality (Sierra-Sosa et al., 2021).

## 5.2 Data Encoding

Even though the current quantum resources and methodologies provide close estimations and approximations of real-world systems, they do not always constitute adequate datasets for quantum classification models. The most common approach in the NISQ era is transforming the well-known classical datasets into the corresponding quantum representation and vice versa (Sierra-Sosa et al., 2021). The fundamental step in the quantum processing pipeline is referred to as *state preparation*. It allows us to process data on quantum circuits before applying any specific quantum computation. Figure 28 shows an example of how quantum encoding is used to project data from the classical space to the quantum (Hilbert) space.

For tackling the data constraints in QML implementations, data encoding is a fundamental step for representing classical data as quantum states. Encoding layers largely influence the QML model expressivity since the data encoding strategy defines and drives the relevant QML parameters, e.g., the features the QML model can represent (Markidis, 2023). Knowing the various encoding techniques is necessary to choose the most suitable one for solving the particular QML task. Each encoding method has constraints due to the unstable quantum mechanical properties that hinder the complete data encapsulation in quantum representation. However, despite that, they successfully encode the information into quantum states and have large margins of improvement over time (LaRose & Coyle, 2020; Li et al., 2022; Weigold et al., 2021). Figure 30 shows a generic overview

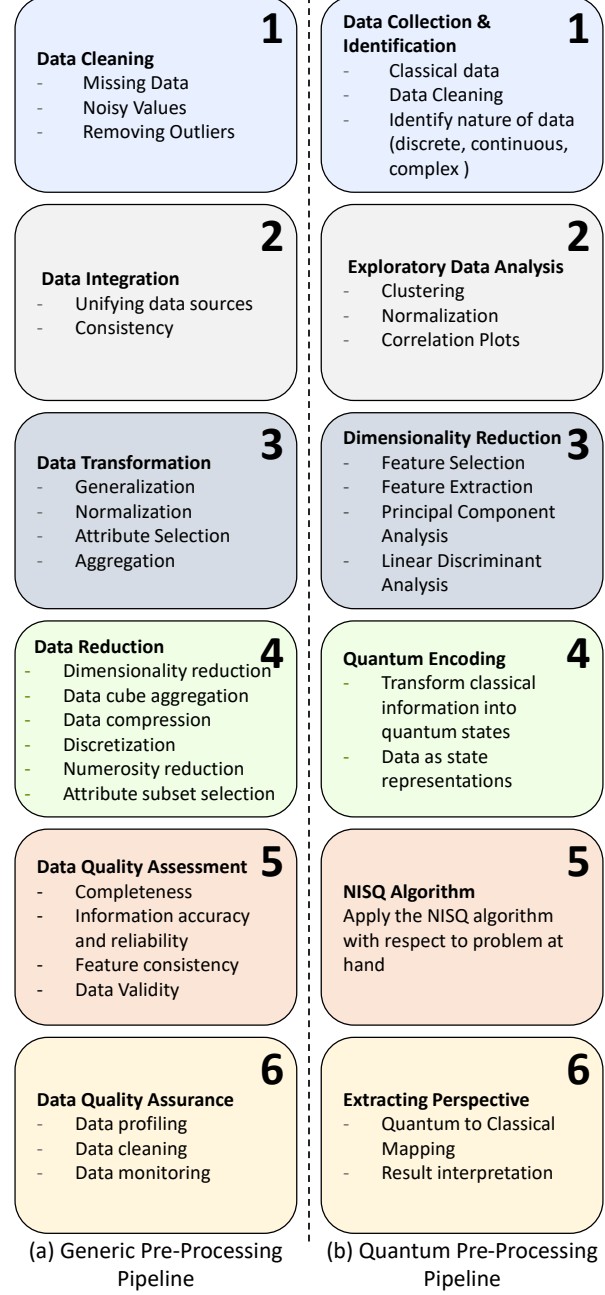

(a) Generic Pre-Processing Pipeline

(b) Quantum Pre-Processing Pipeline

Figure 29: Data pre-processing pipelines. **(a)** Generic pipeline. **(b)** Quantum pipeline.

of the quantum state preparation stage. The process of embedding/encoding generates a quantum representation of the classical data governed by the selected encoding technique within the Hilbert space dimension.

Data encoding patterns describe a particular encoding as a tradeoff between three major objectives:

- The number of qubits needed for the encoding should be minimal because current quantum devices only support a limited number of qubits.

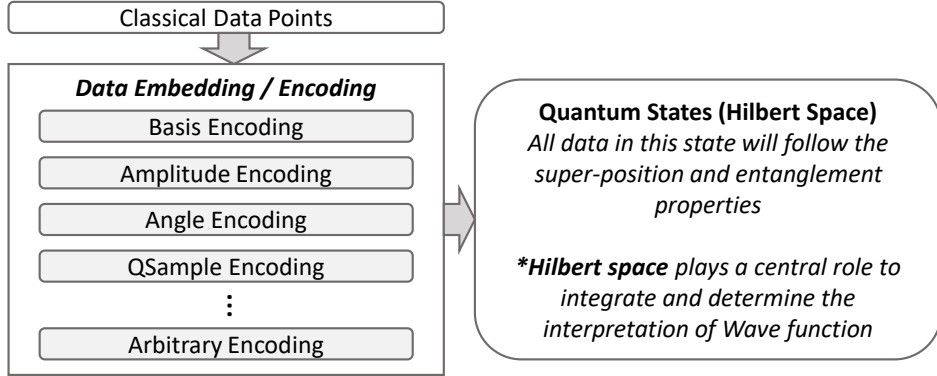

Figure 30: Quantum state preparation model in the data processing pipeline.

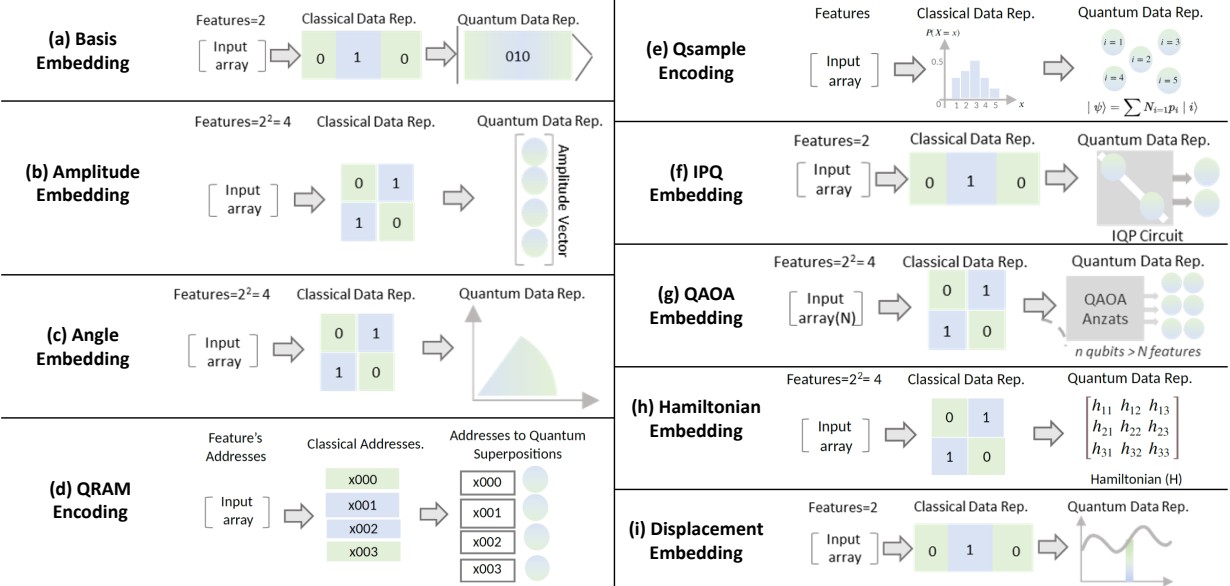

Figure 31: Overview of quantum data encoding techniques. **(a)** Basis Embedding. **(b)** Amplitude Embedding. **(c)** Angle Embedding. **(d)** Quantum Random Access Memory (QRAM) Encoding. **(e)** Qsample Encoding. **(f)** Instantaneous Quantum Polynomial-time (IPQ) Embedding. **(g)** Quantum Approximate Optimization Algorithm (QAOA) Embedding. **(h)** Hamiltonian Embedding. **(i)** Displacement Embedding.

- The number of parallel operations needed to realize the encoding should be minimal to minimize the width of the quantum circuit. Ideally, the loading routine should have constant or logarithmic complexity.

- The data must be represented appropriately for further calculations, e.g., arithmetic operations.

The following list contains the encoding techniques that are used in the literature. Note that the terms *encoding* and *embedding* can be used interchangeably as they refer to the same process. Figure 31 shows an overview of the data encoding techniques and their key characteristics.

(a) The **Basis Embedding** encodes the binary feature vector into a basis state. It is primarily used when real numbers are mathematically manipulated in quantum algorithms. Such an encoding represents real numbers as binary numbers and transforms them into quantum states. However, since the binary features are not differentiable, the basis encoding does not allow gradient computations (Schuld & Petruccione, 2018).

Table 2: Quantum datasets with key features.

| Dataset | Ref. | Type | Volume/Count | Release Date |
|---------|------|------|--------------|--------------|
| QDataSet | (Perrier et al., 2022) | One-qubit and two-qubit systems evolving | 14 TB (compressed), $10,000$ samples each | Aug.2021 |
| QM7, QM8, QM9 | (Rupp et al., 2012) (Montavon et al., 2013) (Ramakrishnan et al., 2015) (Ramakrishnan et al., 2014) | Organic molecules | Between $7,165$ and $133,885$ molecules, with 7-9 heavy atoms | 2012-2015 |
| PennyLane Datasets | (Bergholm et al., 2018) | Organic molecules and spin | 42 geometries of 34 molecules, 4 spin models with 100 configurations | Apr.2022 |
| TensorFlow Quantum Data | (Farhi & Neven, 2018) | Image classification | One-qubit per pixel | Mar.2020 |
| Mendley Datasets for Quantum Circuit Mapping | (Acampora et al., 2021) | Circuit mapping problem as classification task | $188,434$ random quantum circuits | Sep.2021 |
| QFlow Lite | (Zwolak et al., 2018) | Charge state recognition | $1,001$ - $1,599$ measurements, $100 \times 100$ - $250 \times 250$ pixel maps | Sep.2018 |
| NTangled Datasets | (Schatzki et al., 2021) | Quantum states with multipartite entanglement | $12,040$ training samples, $4,060$ test samples | Nov.2021 |

(b) In the **Amplitude Embedding** technique, classical data is encoded into the amplitudes of a quantum state. Besides defining the measurement probabilities, each quantum system's wavefunction can be used to represent its amplitude as a data value. The main advantage of the amplitude embedding is that it can encode $n$ real values (or $n$ fixed-point precision approximation of real numbers) using $\mathcal{O}(\log n)$ qubits, i.e., it has a logarithmic runtime dependency on the dataset size. Due to its efficiency, it is employed in many QML algorithms (Schuld & Petruccione, 2018).

(c) The **Angle Embedding** is performed by applying rotations on the x-axis or y-axis using specialized quantum gates, called Pauli rotation gates, along with the values that must be encoded. If we apply angle embedding on a dataset, the number of rotations is the same as the number of features in the dataset (Schuld & Petruccione, 2018).

(d) The **Quantum Random Access Memory (QRAM) Encoding** (Park et al., 2019) is a mechanism that allows accessing classically stored information in superposition by querying an index register. In other words, it enables direct access to the quantum data given the corresponding classical data value. By introducing a conditional rotation and branch selection procedure, we can feed the QRAM with the classical address of the required quantum data and get the qubit state at the output.

(e) The **Qsample Encoding** associates a real amplitude vector with a classical discrete probability distribution. It can be considered a hybrid case of Basis and Amplitude Encoding because amplitudes represent the encoded information, but the features are encoded in the qubits (Schuld & Petruccione, 2018).

(f) The **Instantaneous Quantum Polynomial-time (IQP) Embedding** (Havlíček et al., 2019) employs the so-called IQP circuit, which is a quantum circuit of a block of Hadamard gates, followed by a set of gates that are diagonal in the computational basis. Each diagonal gate is composed of a single-qubit RZ rotation gate, encoding the $n$ features, followed by a two-qubit ZZ entangler. The entangler, whose pattern can be customized, encodes the product between features.

(g) The **Quantum Approximate Optimization Algorithm (QAOA) Embedding** (Farhi et al., 2014) encodes $n$ features into $m$ qubits, where $m > n$. It uses a layered trainable quantum circuit inspired by the QAOA Ansatz. The feature-encoding circuit associates features with the angles of RX rotations. The main advantage is that it supports gradient computations with respect to both the features and the weights.

(h) The **Hamiltonian Embedding** scheme encodes features into the values of the Hamiltonian operator matrix (Schuld & Petruccione, 2018). The Hamiltonian operator of a system is a quantum mechanical operator that corresponds to the total energy of the system. Unlike previous methods that encode the data in the qubits' states, the Hamiltonian embedding method encodes the data into the operator. Since

it is highly correlated with physics, various physical QML algorithms, including VQE and QAOA, are developed using the Hamiltonian encoding method.

(i) The **Displacement Embedding** encodes $n$ features into the displacement of amplitudes or phases of $m$ modes, where $n \leq m$. It is used in continuous-variable quantum computing models, where classical information is encoded in the displacement operator parameters (Markidis, 2023).

### 5.3 Quantum-Native Data

In quantum mechanics, the fundamental principle is the evolution of energy systems over time, i.e., the Hamiltonian of the systems. In the domain of quantum computing, we deal with varying states of the system for computation and processing. Hence, quantum computing systems must be capable of handling data dictated by these transformations, mechanics, and quantum constraints. *Quantum data* can be referred to as relevant information describing the states and evolution of the quantum system over time. These quantum features represent input and output arguments for various quantum operations and functions that transform the system's quantum state from one form to another. The collection of such data is referred to as *quantum datasets*.

Some examples of quantum data can be:

- Factors affecting the relevant characteristics of a quantum system such as the Hamiltonian and other observables.

- A particular stationary state of the quantum system (e.g., the ground state), from where the process of evolution begins encapsulated in a quantum computation process.

- Unitary transformations, the possible quantum operations applicable to the system.

- Result of computation attained by measurement or additional projection operators which allow extraction of resulting distributions and expectation values of the system.

- Noise-relevant data for the systems' evolution and other phenomena collected via the control parameters.

The data collected for solving real-life problems are currently in classical form. To use quantum models for these solutions, we need to understand and process data in its quantum equivalent form. The possible types of classical data required to create quantum state representations for quantum processing are the following:

- Discrete data (binary or integer values).

- Real continuous data (floating-point values).

- Complex continuous data (complex values).

While fully quantum datasets will drive the FTQC era, the goal in the NISQ era is to understand how to encapsulate the current classical information into a quantum state. In this regard, after understanding the quantum pre-processing, we discuss encoding and embedding techniques for classical data into quantum data. Then, we provide a curated list of quantum datasets along with their properties and application domains.

### 5.4 Motivation for QML Datasets

A valuable lesson from the success of classical ML is that easily accessible and high-quality datasets catalyze the development of new algorithms and the improvement of existing ones. Learning from quantum data is more efficient when executed on quantum computers. Hence, deploying ML algorithms on quantum computers promises efficiency, accuracy, and rich data encapsulation into computing information about real-world systems.

Readily available data drives the efficiency of research quality. High-quality data curated for research and development catalyzes practical and collaborative research across disciplines and fosters insightful growth. Similarly, to experience the full potential of quantum computers to extract information and deeply understand a system through accurate simulations of its real-world counterpart, it is important to have quantum data available to diverse users.

By aligning dataset choice with the intended QML task and the available hardware (qubit count, coherence), researchers can fairly benchmark quantum models against classical baselines and progressively build toward demonstrations of quantum advantage.

## 5.5 Quantum Datasets Collection

Selecting appropriate datasets for QML is a crucial step that impacts both algorithm design and performance benchmarks. Key considerations include the type of data (quantum vs. classical), the feature dimensionality, and the feasibility of encoding data into quantum states given current hardware constraints. Classical datasets (e.g., images or molecule descriptors) are plentiful and often used, but encoding high-dimensional classical information onto qubits can require significant overhead (one qubit per feature or complex amplitude encoding) and careful embedding schemes. By contrast, quantum-native datasets, i.e., data originating from quantum states or simulations, may better exploit QML's strengths. Researchers must balance dataset complexity with quantum hardware limits, often resorting to downscaled or synthetic data to fit the available qubit count and circuit depths. Typical benchmark tasks in QML include classification (e.g., distinguishing quantum states or classifying data points), regression (e.g., predicting molecular energies), Hamiltonian learning (inferring system Hamiltonians or optimal controls from data), and state discrimination (distinguishing or identifying quantum states). The choice of dataset should align with the target task: for instance, quantum chemistry datasets enable testing QML for property prediction, whereas quantum state datasets probe a model's ability to learn inherently quantum patterns. In all cases, benchmarking on well-defined datasets helps compare quantum vs. classical approaches and track quantum advantage. We note that developing standardized quantum datasets (with easily preparable states and known properties) is an active effort to facilitate fair and relevant QML evaluations(Schatzki et al., 2021; Huang et al., 2022).

Table 2 shows the collection of quantum datasets with key features. The following list provides a brief description of each dataset.

- The **QDataSet** (Perrier et al., 2022), released in 2022, is composed of 52 datasets based on simulations of one-qubit and two-qubit systems evolving in the presence or absence of noise. It provides a large-scale set of datasets for QML practitioners to train, benchmark, and develop classical and quantum algorithms for common tasks in quantum sciences, such as quantum control, quantum tomography, and noise spectroscopy. It has been generated using customized code running on base-level Python packages to facilitate interoperability and portability across common QML platforms. Each dataset consists of $10,000$ samples and includes a range of information (stored in list, matrix, or tensor format) regarding quantum systems and their evolution, such as quantum state vectors, drift and control Hamiltonians, unitaries, Pauli measurement distributions, time series data, pulse sequence data for square and Gaussian pulses, noise and distortion data. The total compressed size of the QDataSet (compressed with Pickle and zip) is around 14 TB, while its uncompressed size is around 100 TB. Researchers use QDataSet to train and evaluate models on tasks like quantum control (designing pulse sequences to reach target states), quantum tomography (reconstructing quantum states from measurement data), and noise spectroscopy (learning noise parameters). A major advantage is its breadth for testing algorithms' scalability, but its sheer size (terabyte-scale) poses practical loading and memory challenges. Encoding this data into quantum circuits is non-trivial, since each sample may represent a quantum state or time series that would require many qubits or time steps to embed. Thus, QDataSet is often used in classical-quantum hybrid experiments (or simulations) where classical ML models or quantum-inspired algorithms ingest the data, rather than directly preparing each sample on quantum hardware. Nevertheless, it serves as an important benchmark for validating QML methods on tasks intrinsic to quantum systems (e.g., learning system Hamiltonians or optimal controls) in a controlled setting.

- The **QM7, QM8, and QM9** datasets are subsets of the GDB-13 (Blum & Reymond, 2009) and GDB-17 (Ruddigkeit et al., 2012) databases, which contain billions of organic molecules. The QM7 (Rupp et al., 2012) is composed of $7,165$ molecules of up to 23 atoms (including 7 heavy atoms such as C, N, O, and S), represented using the Coulomb matrix. An extension of the QM7 dataset for multitask learning, called QM7B (Montavon et al., 2013), has 13 additional properties (e.g., polarizability, HOMO and LUMO eigenvalues, excitation energies) and contains a total of $7,211$ molecules. The QMB8 (Ramakrishnan et al., 2015) has a training set of $10,000$ molecules with up to 8 heavy atoms. The QM9 (Ramakrishnan et al., 2014) contains $133,885$ molecules with up to 9 heavy atoms. In QML, these datasets serve as representative regression benchmarks for molecular properties (e.g., predicting atomization energies. Hybrid quantum-classical models (such as variational quantum algorithms or quantum kernel methods) have been tested on subsets of QM7/QM8/QM9 to evaluate their performance relative to classical models in chemistry tasks. For example, a quantum kernel regression approach was shown to predict QM9 molecular energies with competitive accuracy, hinting that quantum feature maps can capture molecular structure–property relationships (Chang, 2022). However, these datasets pose encoding challenges: each molecule must be described by high-dimensional features. To input them into a quantum model, techniques like dimensionality reduction or efficient feature encoding are required to avoid using hundreds of qubits. The relatively large dataset sizes also mean that current QML experiments typically train on smaller samples or simplified tasks. Even so, QM7/8/9 provide valuable testbeds for QML models aiming to learn complex quantum-mechanical relationships in chemistry, and they are often used to benchmark quantum regression algorithms or to assess quantum models' ability to generalize from limited training data.

- The **PennyLane Datasets** (Bergholm et al., 2018) are composed of the *Quantum Chemistry (QChem) Datasets* for common molecular systems and the *Quantum Many-Body Physics (QMBP) Datasets*. The QChem datasets contain the electronic structure data for 42 different geometries of molecules such as linear hydrogen chains, metallic and non-metallic hydrides, and charged species. For each geometry of the molecule, it is possible to extract molecular data (i.e., information regarding the molecule), Hamiltonian data (i.e., Hamiltonian for the molecular system), tapering data (i.e., features based on Z2 symmetries of the molecular Hamiltonian for performing tapering), tapered observables data, and variational data. The QMBP datasets contain spin models displaying quantum correlations. For each spin system, datasets are generated for 1-D lattices (linear chain) and 2-D lattices (rectangular grid). Each dataset contains values for 100 different configurations of tunable parameters such as the external magnetic field and coupling constants. The PennyLane datasets are commonly used to benchmark variational quantum algorithms and QNNs on physically motivated problems. The QChem datasets provide molecular Hamiltonians and ground state energies, supporting studies where quantum models, such as VQEs, aim to predict molecular energy landscapes across different geometries (Romero et al., 2018; Huggins et al., 2021). The QMBP datasets offer spin system data suitable for phase recognition and classification, allowing quantum classifiers to identify ordered and disordered phases based on Hamiltonian configurations (Cong et al., 2019). Since these datasets are expressed in quantum-native formats (Pauli operators, energy spectra), they can be integrated naturally into quantum circuits. However, encoding continuous variables, such as coupling strengths or bond lengths, remains challenging and often requires data re-uploading techniques (Pérez-Salinas et al., 2020).

- The **TensorFlow Quantum Data** builds upon classical datasets such as MNIST (LeCun et al., 1998) and Fashion-MNIST (Xiao et al., 2017). It implements the method proposed by Farhi & Neven (2018) to use binary encoding for representing each pixel with a qubit, with the state depending on the pixel's value. These classification tasks allow direct comparisons between quantum models and classical deep networks on the same data, exploring whether quantum models can achieve similar accuracy with fewer parameters or different scaling. A key challenge here is the scaling of encoding: encoding even moderate-size images quickly becomes intractable. For instance, a full $28 \times 28$ image would require 784 qubits in a straightforward pixel-to-qubit scheme like angle embedding. As a result, researchers either restrict to very low-resolution images or use feature extraction, like Principal Component Analysis (PCA) as in some QML benchmarks (Yu et al., 2019) to reduce dimensionality. While proof-of-concept results like Huang et al. (2021) have shown that quantum models can learn from such data, a concrete quantum advantage may only emerge in the fault-tolerant quantum computing era.

- The **Mendley Datasets for Quantum Circuits Mapping** (Acampora et al., 2021) helps to solve the circuit mapping problem as a classification task in a supervised learning setting. Each dataset contains features related to the calibration data of the physical device and others related to the generated quantum circuit for specific IBM quantum machines, such as IBMQ Santiago, IBMQ Athens, and IBMQ 16 Melbourne. By training on these features and labels, classical ML models have been developed to predict good qubit mappings for new circuits, effectively learning to optimize circuit placement without running an exhaustive search. In the context of QML, it is also possible to train a quantum model (e.g., a variational classifier) on this data to evaluate whether a quantum learner can assist in compiler optimization.

- The **QFlow Lite** dataset (Zwolak et al., 2018) contains a Python-based software suite to train neural networks to recognize the state of a device and differentiate between states in experimental data. It consists of $1,001$ idealized simulated measurements with gate configurations sampling (stored as $100 \times 100$ pixel maps) over different realizations of the same type of device. The QFlow Lite represents a reference dataset for researchers to use in their experiments for developing QML approaches and concepts. The expanded dataset, denominated QFlow 2.0, consists of $1,599$ idealized simulated measurements stored as $250 \times 250$ pixel maps. In addition, the QFlow 2.0 dataset includes two sets of noisy simulated measurements, one with a noise level varied around 1.5 times the optimized noise level, and the other with a noise level ranging from 0 to 7 times the optimized noise level. For QML research, QFlow Lite offers a chance to test quantum models on tasks of state discrimination in a setting that mimics real quantum sensor data: a quantum model could be trained to recognize charge configurations from the pixelated sensor output. The encoding challenge here is substantial because raw images (often $100 \times 100$ or more pixels) would require hundreds of qubits if encoded pixel-by-pixel. Thus, practical QML studies might use downsampled images or extract key features (such as sensor line cuts or PCA components) to feed into a quantum circuit.

- The **NTangled Datasets** (Schatzki et al., 2021) consist of trained weights for three hardware-efficient Ansatzes), strongly entangling, and convolutional, varying the number of qubits, depths, and types of multipartite entanglement. The states are generated by sending product state inputs into PQCs. This resource is explicitly used to benchmark quantum classifiers and QNNs on tasks that involve quantum data. A typical use case is a state classification problem: given measurement data (or the parameter vector) from an unknown state, the task is to determine which entanglement class the state belongs to (for instance, distinguish between product states, GHZ-type entangled states, W-type states, etc.). In the NTangled study, the authors trained QML models (like variational quantum classifiers) on these state datasets and evaluated their accuracy in recognizing entanglement patterns (Schatzki et al., 2021). A big advantage of NTangled is that the data is inherently quantum (provided as quantum circuits or state vectors), allowing quantum models to potentially handle it more naturally than classical models. However, using NTangled on real hardware would require preparing those quantum states on a quantum computer and then running the learning algorithm, which is resource-intensive. Overall, NTangled's primary use is to test and improve QML methods for entanglement recognition and classification, showcasing the promise of quantum-enhanced learning on data that classical methods find challenging.

## 6 Tools and Technologies

This section introduces the tools, simulators, and hardware models that are available for QML practitioners. Since the infrastructure provided by QML industries is rapidly evolving, it is important to keep updated about the new technologies and software.

### 6.1 Quantum Hardware

Single qubits can be implemented using different materials and principles. As shown in Figure 32, companies do not use the same technology for implementing their quantum computers. For example, IBM Quantum uses niobium and aluminum on a silicon base to achieve superconductivity under near absolute zero temperature (IBM, 2022). IonQ uses trapped ion quantum made of Ytterbium, a rare metal (IonQ). Another example is Xanadu, which applies photonics (light particles) in the quantum processing unit (Madsen et al., 2022). Using ion and light has advantages over the superconductivity materials as it works under room

temperature. The superconducting qubits are more scalable and are implemented by other companies like Google (Frank Arute, 2019) and D-Wave (McGeoch & Farre, 2020). It is exciting to see future innovations in this area. In this section, after discussing the set of conditions necessary for designing qubits (denoted as DiVincenzo's criteria), we review the qubit technologies to understand their advantages and disadvantages and the reasons for their adoption in the industry. A detailed view of the quantum hardware technologies is also illustrated in Table 3.

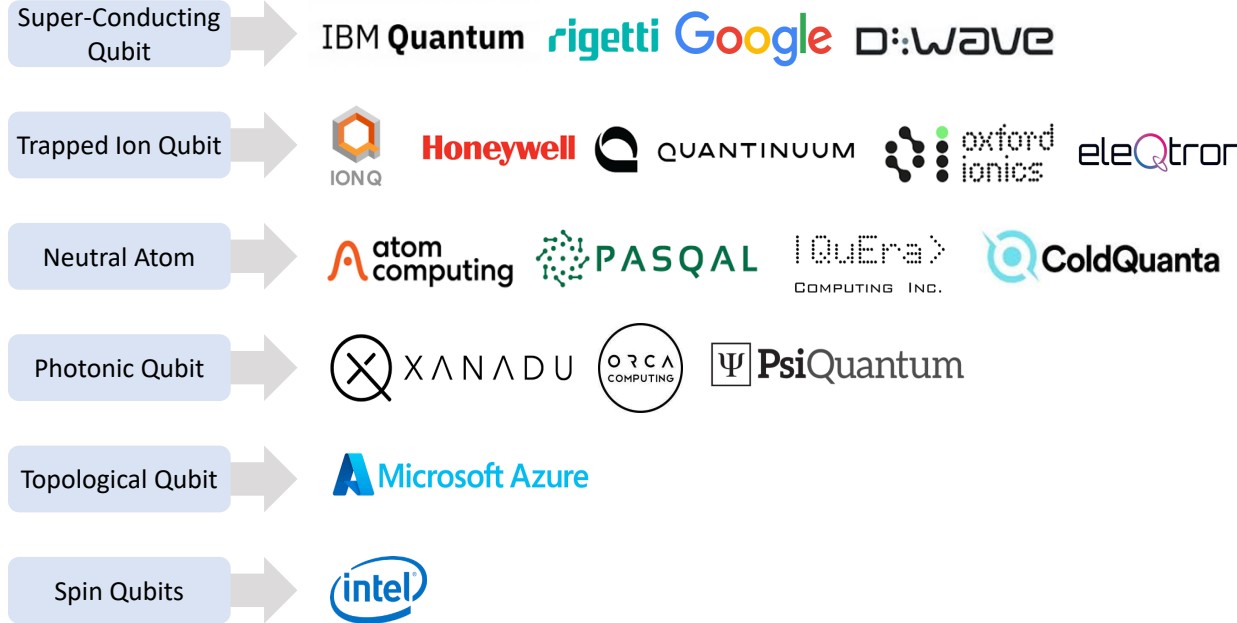

Figure 32: Qubit technologies used by companies for their quantum computers.

### 6.1.1 DiVincenzo's Criteria

Introduced by theoretical physicist David P. DiVincenzo in the year 2000, DiVincenzo's criteria (DiVincenzo, 2000) defines a set of conditions necessary for constructing a quantum computer. The community widely accepts it, and, in the current state of the field, it represents a useful tool for designing qubits. The following list contains the fundamental characteristics for designing new qubit modalities in suitable experimental setup.

- Scalability with well-characterized qubits

- Ability to initialize qubits to a simple fiducial state

- Long relevant decoherence times

- A "universal" set of quantum gates

- A qubit-specific measurement capability

While the above criteria are necessary for designing qubits, the following two criteria are the minimum fundamental characteristics required for quantum communication.

- Ability to interconvert stationary and flying qubits

- Ability to faithfully transmit flying qubits between specified locations

Table 3: Quantum hardware technologies with key features.

| Brand & Technology | Processor | Ref. | Connectivity | # Qubits | Single-Qubit Native Gates | Multi-Qubit Native Gates | Release Date |
|---|---|---|---|---|---|---|---|
| **IBM** Superconducting Qubit | Osprey | (IBM, 2022) | All-to-All | 433 | ID, RZ , SX, X | Echoed cross resonance gate (ECR) | Nov.2022 |
| | Eagle | | | 127 | | | Nov.2021 |
| | Egret | (IBM) | | 33 | | CZ | Apr.2022 |
| | Hummingbird | | | 65 | | CX | Nov.2020 |
| | Falcon | | | 27 | | | Nov.2019 |
| | Canary | | | 5-16 | | | Jan.2017 |
| **IonQ** Trapped Ions | Forte | (Chen et al., 2023) | All-to-All | 32 | Gpi, GPi2, Virtual Z | Fully Entangling MS, Partially Entangling MS | 2022 |
| | Aria | (IonQ) | | 21 | | | 2022 |
| | Harmony | (IonQ) | | 9 | | | 2020 |
| **Xanadu** Photonic Qubits | Borealis | (Madsen et al., 2022) | 3D connectivity | 216 | NA | NA | Jun.2022 |
| | X-Series | (Arrazola et al., 2021) | NA | NA | | | Mar.2021 |
| **Google** Superconducting Qubits | Sycamore (Weber QC) | (Frank Arute, 2019) | Square-Lattice Grid | 54 | Phased XZ, Virtual Z, Physical Z, Pauli Gates XYZ, PhasedXPowGate | Sycamore Gate, Square Root of iSWAP, CZ, FSim Gateset, Wait Gate, Parameterized Gates | Oct.2019 |
| | Bristlecone | (Google, 2018) | 9-qubit linear array | 72 | NA | NA | Mar.2018 |
| | Foxtail | (Goo, 2016) | | NA | | | 2016 |
| **Rigetti** Superconducting Qubits | Aspen-M-3 | (Computing) | Octagonal with 3-fold connectivity (2-fold for edges) | 80 | RX (angle, qubit) RZ (angle, qubit) | CPHASE, MEASURE (qubit, classical reg), CZ | Dec.2022 |
| | Aspen-M-2 | | | 80 | | | Aug.2022 |
| | Aspen-M-1 | | | 80 | | | Feb.2022 |
| | Aspen-11 | | | 40 | | | Dec.2021 |
| | Aspen-10 | | | 32 | | | Nov.2021 |
| | Aspen-9 | | | 32 | | | Feb.2021 |
| | Aspen-8 | | | 31 | | | May.2020 |
| | Aspen-7 | | | 28 | | | Nov.2019 |
| | Aspen-4 | | | 13 | | | Mar.2019 |
| | Aspen-1 | | | 16 | | | Nov.2018 |
| | Acorn | | | 19 | | | Dec.2017 |
| | Agave | | | 8 | | | Jun.2017 |
| **Intel** Spin Qubits | Intel Quantum Hardware | (Intel) | NA | 12 | NA | NA | 2023 |
| **Quantinuum (Honeywell)** Trapped Ions | H2-1 | (Quantinuum) | fully connected qubits | 32 | Rotation Gates | ZZ | Aug.2023 |
| | H1-3 | | | 20 | | | Jun.2022 |
| | H1-2 | | | 12 | | | Jul.2021 |
| | H-1 | | | 10 | | | Sep.2020 |
| **Atom Computing** Neutral Atoms | Phoenix | (Barnes et al., 2022) | 8:1 connectivity | 100 | NA | NA | Jul.2021 |
| **QuEra** Neutral Atoms | Aquila | (Wurtz et al., 2023) | NA | 256 | Rabi Oscillations Time-dependent protocols Dynamic Decoupling Protocols | Adiabatic State Preparation and Rydberg Blockade Rabi Frequency Enhancement Levin-Pichler gate analogues Non-Equlibrium dynamics of 2 qubits | Jul.2021 |
| **D-Wave Systems** Superconducting Qubits | Advantage | (McGeoch & Farre, 2020) | Degree-15 Lattice $(15 \times 15 \times 12)$ | 5,000+ | superconducting quantum annealing | superconducting quantum annealing | Sep.2022 |
| | 2000Q | (King et al., 2017) | Degree-6 Lattice $(8 \times 8 \times 8)$ | 2,000+ | Quantum Annealing | Quantum Annealing | Mar.2017 |

**Legend ID:** Identity. **RZ:** Rotation around Z-axis. **X:** NOT gate. **SX:** Square root of X. **CZ:** Controlled Z-gate. **CX:** Controlled X-gate. **Gpi:** π or bit-flip rotation with an embedded phase. **GPi2:** $RX\left(\frac{\pi}{2}\right)$ or $RY\left(\frac{\pi}{2}\right)$ with an embedded phase. **Fully Entangling MS:** XX gate, a simultaneous, entangling $\frac{\pi}{2}$ rotation on both qubits. **Partially Entangling MS:** a third (optional) arbitrary angle θ is added. **FSim Gateset:** provides all three Sycamore, iSWAP, and CZ gates in one set. In addition, by using this combined gate set, it can be parametrized, which allows for efficient sweeps across varying two-qubit gates. **CPHASE:** operation applied with a rotation parameter θ. **CZ (Rigetti):** offers single-pulse construction, allowing for lower error rates when full phase control is not needed.

## 6.1.2 Superconducting Qubits

Superconducting circuits are macroscopic in size but detain generic quantum properties such as entanglement, quantized energy levels, and superposition of states, all of which are more commonly associated with atoms. Their quantum state is manipulated using electromagnetic pulses to control the electric charge, the magnetic flux, or the phase difference across a Josephson junction (Kockum & Nori, 2019) (a device with nonlinear inductance and no energy dissipation). As such, superconducting qubits represent the primitive building blocks of quantum computers (Bravyi et al., 2022). There exist multiple types of superconducting qubits, such as flux qubits, phase qubits, charge qubits, and Transmon qubits. The most popular type is the Transmon qubit (Salmanogli, 2021) given its design capability to limit noise effects, supporting error mitigation and correction.

## 6.1.3 Semiconducting Quantum Dots & Spin Qubits

Semiconductor charge and spin qubits based on gate-controlled semiconductor quantum dots, shallow dopants (i.e., traces of an impurity element introduced into a chemical material to alter its original electrical or optical

properties), and color centers in wide-bandgap materials. Semiconductor Quantum Dots (QDs) (Harvey, 2022) are nanoscale material clusters composed of 102–105 atoms. The size of the QDs is orders of magnitude larger than a typical atomic radius, yet small enough to provide quantum confinement of electrons and holes in all three spatial dimensions. Depending on their configuration, they can be controlled by both magnetic and electric fields. Therefore, they can be dephased by electric and magnetic field noise, with different types of spin qubits having various control mechanisms and noise susceptibilities.

In semiconductor quantum dots, spin qubits represent a prominent class of solid-state qubits in the effort to build a quantum computer. The simplest spin qubit is a single electron spin located in a quantum dot. However, many additional varieties have been designed, some containing multiple spins in multiple quantum dots, each with different benefits and drawbacks. Spin qubits based on solid-state defects have emerged as promising candidates because these qubits can be initialized, selectively controlled, and read with high fidelity at ambient temperatures.

### 6.1.4 Trapped Ions

As the name suggests, the qubits are ions trapped by electric fields and manipulated with lasers (Nielsen & Chuang, 2010a). Trapped ions have relatively long coherence times, which implies that the qubits are long-lived. Moreover, they can easily interact with their neighbors.

### 6.1.5 Neutral Atoms

In the neutral atom quantum computing model, light beams manipulate arrays of single neutral atoms to encode and measure quantum states. In these quantum processors, a qubit is defined by one of two electronic states of an atom, and these single neutral atoms are arranged in configurable arrays. Neutral atom hyperfine qubits provide inherent scalability owing to their identical characteristics, long coherence times, and the ability to be trapped in dense, multidimensional arrays (Henriet et al., 2020). Combined with the strong entangling interactions provided by Rydberg states (Shi, 2022), all the necessary characteristics for quantum computation are met.

### 6.1.6 Photonic Atoms

One way to get scalable structures is to use photons (i.e., light particles) to represent qubits (Slussarenko & Pryde, 2019). It consists of a ring that is used for photon storage, along with the scattering unit. Unlike other physical systems, photonics allows us access to an infinite number of states.

### 6.1.7 Transition Metal Dichalcogenides

Among the possibilities in the solid state, a defect in Transition Metal Dichalcogenides (TMDs), known as the nitrogen-vacancy (NV-1) center, stands out for its robustness. Its quantum state may be initialized, manipulated, and measured with high fidelity at room temperature. These controls can be used in conjunction with electronic structure theory to smartly sort through candidate defect systems for probable qubit representation (Tsai et al., 2022). However, this technology has not really been popular in the industry until now.

### 6.1.8 Topological Nanowire Qubits (Majorana Qubits)

Majorana particles are a unique class of topological nanowire particles that are their own antiparticles. The quasi-particles emerge from the topological nature of systems and are bound at zero energy. They are predicted to obey non-abelian braiding statistics such that they can "remember" their histories. A *pair* of Majorana fermions represents a single qubit, and its realization could lead the way to fault-tolerant quantum computing (Tutschku et al., 2020). Despite some initial hurdles (Frolov, 2021; Legg, 2025), Majorana Qubits are extremely promising due to their stability at close to normal temperatures, and are the technology chosen by Microsoft (Aghaee et al., 2023; Aasen et al., 2025).

### 6.1.9 Pros and Cons of Leading Quantum Technologies

**Superconducting Qubits:** *Pros:* Fast gate speeds (nanoseconds) and high gate repetition rates, with solid-state fabrication methods that have already scaled to hundreds of qubits (demonstrated in chips by Google, IBM, etc.), showing compatibility with integrated circuit techniques for scalability. *Cons:* Relatively short coherence times (typically tens of microseconds up to a few hundred microseconds at best), meaning qubits quickly lose quantum information, and noise from materials and control electronics leads to error rates on the order of $10^{-3}$–$10^{-4}$ per gate (Kjaergaard et al., 2020). They also require dilution refrigerators operating at millikelvin temperatures for stability, incurring significant overhead in operational temperature and complexity.

**Trapped Ion Qubits:** *Pros:* Individual ions confined in electromagnetic traps have very long coherence times (on the order of seconds or more, as the ions' internal states are well-isolated from the environment). They achieve high-fidelity gates (single- and two-qubit gate errors as low as $10^{-3}$ or better) and all-to-all connectivity within a trap, making noise resilience and entangling operations robust (Bruzewicz et al., 2019). Trapped ions operate in ultra-high vacuum but essentially at room temperature (no cryogenics needed, only laser cooling), greatly easing the operational requirements. *Cons:* Gate speeds are much slower (two-qubit gate durations in the millisecond range), limiting the processing speed. Scaling to large numbers of qubits in one device is challenging. Ion traps with more than 50 qubits face mode crowding and crosstalk, and while modular architectures (networking multiple traps) are proposed, they add complexity. Thus, scalability is moderate, since trapped-ion systems excel in qubit quality but require significant engineering (laser systems, stable trapping) to handle many qubits and fast operations.

**Neutral Atom Qubits:** *Pros:* Neutral atoms (e.g., atoms in optical tweezers) offer inherently scalable architectures. Hundreds of atoms can be arranged in 1D or 2D arrays with each atom as an identical qubit. They have long coherence times using stable hyperfine states (similar to trapped ions), and use highly excited Rydberg states to induce strong, controllable interactions for two-qubit gates (enabling entanglement across neighboring atoms). Neutral atom platforms operate at near-room temperature (cold atoms are trapped by lasers in vacuum, but no dilution fridge is required). *Cons:* Two-qubit gate fidelities, while improving, are currently lower than in ions or superconductors (errors on the order of a few percent), due to relative decoherence of short-lived Rydberg excitations and sensitivity to laser phase noise (Henriet et al., 2020; Shi, 2022). Achieving uniform control laser focus and stability across a large array is technically complex, which can impact scalability even though the physical layout is extensible. Additionally, gate speeds for Rydberg-mediated operations are intermediate (microsecond scale), so while faster than ions, some algorithms still face time limitations given qubit coherence constraints. Overall, neutral atoms promise high qubit counts and reconfigurability, but require further advances in control precision and error correction to match the gate fidelities of other technologies.

**Photonic Qubits:** *Pros:* Photonic quantum computing uses photons (light) as qubits, which naturally suffer virtually no decoherence, because a photon traveling through space maintains its quantum state until interaction or loss, giving effectively infinite coherence in transit. Photonic systems can operate at ambient conditions (operational temperature is typically room temperature), and are ideal for communication (flying qubits) and networking of quantum information. They also offer large Hilbert spaces (e.g., encoding in light modes allows many-state qubit generalizations). *Cons:* Direct two-qubit interactions between photons are difficult; operations often rely on probabilistic gates or effective interactions mediated by matter, which significantly complicates scalability for quantum computing purposes. Scaling photonic qubit counts and circuits is challenging due to losses. As the number of components (beam splitters, phase shifters, detectors) grows, photon loss and detection inefficiency cause error rates to rise. Noise resilience in terms of decoherence is high, but photon loss acts as a dominant noise source. Moreover, while photons do not need cooling to preserve state, high-efficiency single-photon sources and detectors often require cryogenic technology, adding to the experimental overhead (Slussarenko & Pryde, 2019). In summary, photonic qubits excel in coherence and connectivity (they can be sent over long distances), but the difficulty of implementing deterministic entangling gates and the challenges in reliably generating and detecting large numbers of photons make building large-scale photonic quantum computers an ongoing research effort.

Table 4: Quantum simulators with key features.

| Brand | Simulator Name | Type | # Qubits | Noise Model |
|---|---|---|---|---|
| **IBM** | simulator_statevector | Schrödinger wavefunction | 32 | Yes |
| | simulator_stabilizer | Clifford | 5000 | Yes (Clifford only) |
| | simulator_extended_stabilizer | Extended Clifford | 63 | No |
| | simulator_mps | Matrix Product State | 100 | No |
| | ibmq_qasm_simulator | General, Context-Aware | 32 | Yes |
| | sampler | Qiskit Runtime Primitive (quasi-probability dist. generator) | NA (inherited from backend) | Yes (via *options* parameter) |
| | estimator | Qiskit Runtime Primitive (expectation values calculator) | NA | Yes (via *options* parameter) |
| **Xanadu** | default.qubit | Qubit Simulator | up to 26 | Yes, non-native |
| | default.mixed | Qubit Simulator | up to 16 | Yes (Native support) |
| | default.gaussian | Qubit Simulator | NA | Yes, non-native |
| | lightning.qubit | Quantum Photonic Simulator | up to 26 | Yes, non-native |
| | lightning.gpu | GPU-Accelerated Simulator | up to 29 (+3 with MPI optimization) | Yes, non-native |
| | lightning.kokkos | Plugin for Kokkos Library | up to 29 (+3 with MPI optimization) | Yes, non-native |
| | Jet | tensor network contractions (acceleration) | NA | Yes, non-native |
| **Rigetti** | Quantum Virtual Machine (QVM) | Emulated Execution | NA | Yes (built-in and custom) |
| **Intel** | Intel Quantum Simulator (IQS) | Generic Qubit Simulator | 32, 40 | NA |
| **Quantinuum (Honeywell)** | H2 Emulator | Functional Simulator | 32 | Yes (via Quantinuum API) |
| | H1 Emulator | Functional Simulator | 20 | Yes (via Quantinuum API) |
| **Google** | Qsim | Full Wave Function Simulator | 40 | Yes (via *channel* in Criq) |

## 6.2 Quantum Circuit Mapping

The process of circuit mapping consists of compiling a quantum circuit to the given device. The circuit is transformed to comply with the architecture's limited qubit connectivity.

Combining one-qubit rotation gates and two-qubit Controlled-NOT (CNOT) gates can realize any quantum circuit. However, CNOT gates cannot be applied to all pairs of qubits in the currently available NISQ quantum devices. Therefore, the circuit must be transformed into an equivalent circuit that does not violate the limitation of the device. The transformation often requires additional gates that limit the usability of the device (Itoko et al., 2020). The quality of the circuit mapping can be assessed via different metrics, such as the resulting circuit's (two-qubit) gate count, depth, or the expected fidelity (Wille & Burgholzer, 2023).

Circuit mapping is a fundamental step in the context of QML. The factors affecting the development of QML algorithms directly depend on the design and optimization of circuit mapping. For instance, considering a circuit mapping capability of coupling up to 4 qubits significantly limits the depth and size of QNN layers.

## 6.3 Quantum Simulators and Primitives

*Quantum Simulators* are cloud-based classical systems emulating quantum systems. These are useful in the NISQ era as designers fully control the hardware characteristics that enable experiments of fault-tolerant quantum systems. Ideal quantum systems can be simulated for long-term design and development with no noise in the system. Moreover, noise models can be introduced and analyzed to comprehend the intuitive and counterintuitive behaviors of the quantum systems under noise.

*Quantum Primitives* are foundational building blocks for designing and optimizing quantum workloads. They provide options to customize the iteration and execution of programs to maximize the solution quality. A primitive is a set of key language elements that serve as the foundation for a programming language. Every

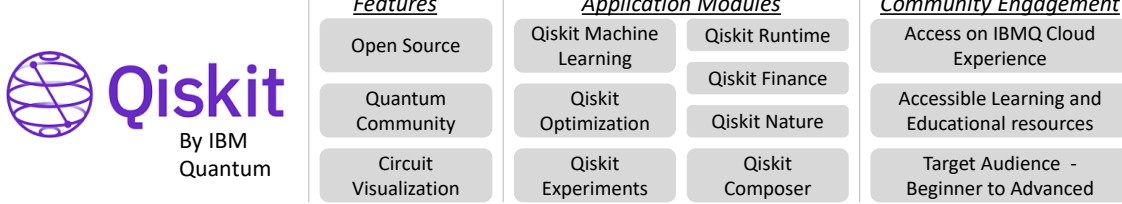

Figure 33: Qiskit features and modules.

language supports a core set of primitives that grant the basic building blocks for instructing a processor on executing specific operations.

To be broadly applicable, quantum computers must accurately control and process the information stored in thousands of quantum bits. However, as the number of qubits increases, noise and cross-talk tend to increase the error rate rapidly. In this thrust, it is important to study how the error rates scale with the increased qubit number and system connectivity. To mitigate this, quantum computers can be built in a modular fashion, with adequate infrastructure needed to control them efficiently, and the software compilation of quantum algorithms should be tailored to specific system architectures and error characteristics.

Several software simulators have been released by major companies involved in QML research. While Table 4 summarizes the key features of the simulators, the following list provides key details of these tools.

- **IBM Qiskit** (Qiskit contributors, 2023) is an open-source software development kit for quantum systems at the level of circuits, pulses, and algorithms. With the help of domain-specific APIs, Qiskit builds a software stack that makes it easy for users to use quantum computers, as it allows practitioners to easily design experiments and applications and run them on classical simulators or real quantum computers. An overview of the main characteristics of Qiskit is illustrated in Figure 33. It comprises the following components:

  - *Qiskit Runtime*: It contains computational primitives to perform foundational quantum computing tasks and supports error correction and mitigation techniques.
  - *Qiskit Composer*: A graphical quantum programming tool that lets users describe the operations to build quantum circuits and run them on real quantum hardware or simulators.
  - *Qiskit Nature*: It supports different applications and provides the necessary components to convert classical codes into representations required by quantum computers.
  - *Qiskit Finance*: It provides a set of illustrative applications and tools, including data providers for real or random data, Ising translators for portfolio optimization, and implementations for pricing different financial options or for credit risk analysis.
  - *Qiskit Machine Learning*: It provides fundamental QK and QNN building blocks for QML algorithms that apply them to solve different tasks such as regression and classification. Moreover, it connects to PyTorch to enhance classical ML workflows with quantum components.
  - *Qiskit Optimization*: It provides a range of optimizations, including high-level modeling of optimization problems, automatic conversion of problems to different required representations, and a suite of easy-to-use quantum optimization algorithms.
  - *Qiskit Experiments*: It runs characterization, calibration, and verification steps for the experiments.
  - *OpenQASM*: It is a simple text-format quantum assembly language for describing acyclic quantum circuits composed of single-qubit, controlled single-qubit, multiple-qubit, and controlled multiple-qubit gates. It is used to implement experiments with low-depth quantum circuits. OpenQASM represents universal physical circuits with straight-line code that includes measurement, reset, fast feedback, and gate subroutines. The simple text language can be written by hand or by higher-level tools and may be executed on the IBM Q Experience.

- **Xanadu PennyLane** (Bergholm et al., 2018) is the leading tool for programming quantum computers. It is a cross-platform Python library that enables quantum differentiable programming and integration

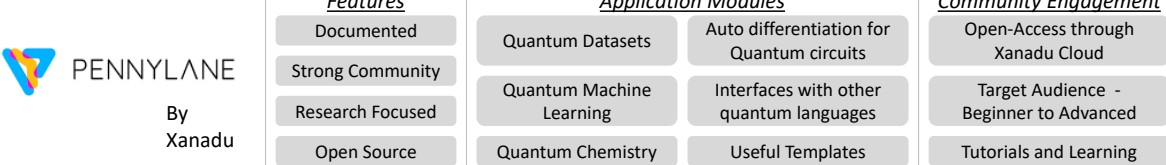

Figure 34: PennyLane features and modules.

with ML tools to train a quantum computer like you would train a DNN. PennyLane also supports a comprehensive set of features, hardware, simulators, and community-led resources that enable users of all levels to build, optimize, and deploy quantum-classical applications easily. An overview of its main features is depicted in Figure 34.

- **Xanadu Strawberry Fields** (Killoran et al., 2019) is a full-stack Python library for designing quantum algorithms and executing them directly on photonic quantum computers such as Xanadu's next-generation quantum hardware.

- **Rigetti PyQuil** (Smith et al., 2016; Karalekas et al., 2020) is a Python library for writing and running quantum programs using Quantum instruction language (Quil) programs. PyQuil allows developers to easily generate Quil programs from quantum gates and classical operations, compile and simulate these programs using the Quil compiler (Quilc) and the Quantum Virtual Machine (QVM), or execute them on a Rigetti quantum processor. Quilc is an optimizing compiler for gate-based quantum programs, capable of performing circuit transformations to produce optimal circuit implementations for a specific Rigetti processor. Such optimizations allow developers to write programs faster while preserving (or improving) their execution fidelity on a given hardware system. Quil is built and executed using the Forest SDK, a software tool that allows users to write quantum programs in Quil.

- **Quantinuum (Honeywell) TKET** (Qua) is an advanced software development kit, accessible through the PyTKET Python package, for the creation and execution of programs for gate-based quantum computers. It is platform-inclusive, and its state-of-the-art circuit optimization routines allow users to extract as much power as possible from any of today's NISQ devices.

- **Google Cirq** (Developers, 2023) is an open-source framework for programming quantum computers, whose software library can be used for writing, manipulating, optimizing, and running quantum circuits on quantum computers and quantum simulators. Cirq provides valuable abstractions for dealing with today's NISQ computers, where hardware details are vital to achieving state-of-the-art results.

- **Google OpenFermion** (McClean et al., 2020) is an open-source library for analyzing and compiling quantum algorithms to simulate fermionic systems, including quantum chemistry. The package provides efficient data structures for representing fermionic operators and fermionic circuit primitives for their execution on quantum devices. Plugins to OpenFermion provide users with efficient and low-overhead means of translating electronic structure calculations into quantum circuit calculations.

- **Google TensorFlow Quantum (TFQ)** (Broughton et al., 2020) is a QML library for rapid prototyping hybrid quantum-classical ML models. Research in quantum algorithms and applications can leverage Google's TFQ, all from within TensorFlow. It integrates quantum computing algorithms and logic designed in Cirq and provides quantum computing primitives compatible with existing TensorFlow APIs, along with high-performance quantum circuit simulators.

- **D-Wave Leap** (DWa) is a programming model and execution framework to execute quantum and hybrid workloads effectively.

### 6.3.1 Comparison of Quantum Software Frameworks

While all the above frameworks enable quantum programming, each has distinct strengths and target use cases. **IBM's Qiskit** is a comprehensive SDK spanning low-level pulse control to high-level algorithms, with

extensive libraries (for chemistry, finance, machine learning, etc.) and a large community. It is well-suited for general-purpose quantum computing across a broad user base, from beginners to advanced researchers (Qiskit contributors, 2023). In contrast, **Google's Cirq** focuses on giving experts fine-grained control over quantum circuits and is tailored to NISQ devices where hardware details matter for performance (e.g., mapping to qubit topology, scheduling gates with minimal error) (Developers, 2023). Cirq is often chosen for research that directly interfaces with hardware (such as calibrations or implementing algorithms on Google's Sycamore processor), whereas Qiskit emphasizes a more hardware-agnostic approach (with cloud access to IBM quantum processors) and a rich set of built-in algorithmic modules. **Rigetti's PyQuil** lies in between: it is designed for writing programs in the Quil assembly language to run on Rigetti's superconducting qubit hardware (or its Quantum Virtual Machine simulator), allowing explicit hybrid quantum-classical computations with shared memory (Smith et al., 2016). This makes PyQuil advantageous for experiments requiring classical feedback within quantum circuits (a feature now emerging in other frameworks as well), though its ecosystem is more tied to Rigetti-specific tools and hardware. Finally, **Xanadu's PennyLane** sets itself apart by targeting QML and differentiable programming. It provides automatic differentiation of quantum circuits and integration with popular ML libraries, enabling optimization of quantum circuits in conjunction with classical neural networks (Bergholm et al., 2018). PennyLane is hardware-agnostic via plugin interfaces, as users can prototype algorithms on simulators or real devices from IBM, Google, Honeywell, etc., through a unified high-level interface.

In summary, Qiskit and Cirq are often favored for developing quantum algorithms and exploring hardware-specific optimizations (with Qiskit offering a broader platform compatibility and Cirq a lower-level control), PyQuil is specialized for co-processing with Rigetti QPUs using a unique ISA, and PennyLane excels in hybrid quantum-classical workflows, particularly for variational algorithms and quantum neural networks. Each framework's design reflects its priorities: Qiskit for completeness and accessibility, Cirq for hardware-centric research, PyQuil for integrating quantum programs with classical computation, and PennyLane for machine-learning-driven quantum experimentation.

### 6.4 Supporting Infrastructure Tools and Hardware

To help users access quantum resources, several cloud-based platforms and development support infrastructures have been built. The following list describes the most popular resources used in the community.

- **qBraid** (qBr) is a cloud-based quantum computing platform designed to accelerate the pace of progress in the field. It offers access to the ecosystem of quantum software, hardware, and an entire suite of application-specific quantum algorithms, helping software engineers run programs on quantum computers available from AWS, IBM, Rigetti, IonQ, and QuEra within minutes. qBraid Lab is a web-based IDE interface (through JupiterLab) that provides software tools for developers and access to quantum hardware. It also includes CLI, a Command Line Interface for interacting with all parts of the qBraid platform.

- **IonQ Quantum Cloud** (Ion) provides access compatible with most Quantum SDKs with their continuous effort to extend the support to newer tools. The cloud provides access to all IonQ's quantum processors and supports noisy and ideal simulators.

- **Amazon Braket by AWS** (Amazon Web Services, 2020) is a fully managed quantum computing service designed to speed up scientific research and software development for quantum computing. Its support is useful for experimenting with quantum computing algorithms, testing different quantum hardware, and building and developing quantum software. It provides access to different quantum computers and simulators using a consistent collection of development tools. Users can build quantum projects on a trusted cloud with affordable pricing and management controls for both quantum and classical workloads. It allows faster execution of hybrid quantum-classical algorithms with priority access to quantum computers and no classical infrastructure to manage.

- **Microsoft Azure Quantum** (Mic) provides quantum cloud access equipped with open-source development toolkits and environments for most leading quantum technologies with runtime metrics.

Microsoft is adopting topological qubits as their qubit modality for developing their quantum computers. Their aim behind using these qubits directly corresponds with the long-term scalability of quantum computers. Since the topological qubits can survive at regular room temperatures, their usage improves the systems' stability compared to other qubit technologies. The resource estimator tool provided by Azure monitors the quantum system's runtime statistics. It estimates the resources allocated by the quantum code and refines the solution to a reasonable extent. This tool calculates the number of logical and physical qubits and runtime requirements to execute quantum applications on future-scaled quantum computers. Additional features, like determining the number of physical qubits required by circuits and evaluating the differences across qubit technologies, facilitate the design of quantum solutions with varying technologies and refining them to scaled versions for FTQC.

- **Zapata Orquestra** (Zap) is a platform to develop and deploy generative AI applications. It allows rapid prototyping, benchmarking, and executing workflows across quantum and classical computing resources. Their objectives focus on prioritizing use cases that can deliver a near-term financial impact and build customer first-mover capabilities, ultimately leading to solving computationally complex problems.

- **Xanadu Cloud** (Xan) is an interface that allows the simulation and execution of programs on quantum photonic hardware. Users can develop quantum applications and program quantum computers with PennyLane's library of tools, demonstrations, tutorials, and community support forums.

## 7 Applications

QC has been pacing itself from theory to practicality over the past decades, with immense efforts and investment attracting its attention, now more than ever. Although, there are significant hurdles in achieving high compute efficiency, a path is still paved in the direction of robust, reliable, and useful quantum computers. There is great potential in solving QML problems in many application areas that are discussed in this section.

In the following domains, the introduction of quantum technology introduces improvements in speed, compatibility, accuracy, or memory that make them suitable to be solved using quantum technologies.

However, demonstrating a real-world quantum advantage in these application domains is contingent on further progress in quantum hardware and on the development of application-specific quantum architectures. In practice, quantum solutions will need more qubits with lower noise and longer coherence times to outperform classical methods on meaningful tasks. Therefore, it is essential to conduct a co-design approach in which quantum algorithms and hardware are engineered together for specific use cases. Such application-specific quantum computing architectures, tuned to particular problem requirements, are key for unlocking practical QML advantages. In other words, with the help of future hardware improvements and specialized quantum algorithm–hardware optimization, QML will make significant strides in real-world applications.

### 7.1 Molecular Simulation: Drug Design

Identifying and developing small molecules and macromolecules that may help cure illnesses and diseases is the primary activity of pharmaceutical companies. Given its focus on molecular formations, the pharma industry is a natural candidate for deploying quantum computing. The molecules (also including those that might be used for drugs) are actually quantum systems, i.e., systems based on quantum physics. Quantum systems can predict and simulate these molecules' structure, properties, and behavior (or reactivity) more effectively than conventional computing, making QML an extensive branch of research in the respective industry. Exact methods are computationally intractable by standard computers, and often approximate methods are not sufficiently accurate for simulating such critical interactions on the sub-atomic level, as is the case for many compounds. As quantum computers become more powerful, significant discoveries will be made.

QC's primary value for pharmaceuticals lies in the R&D phase of drug design (Epifanovsky et al., 2021). QML can help drug developers build predictive models to identify which molecules will likely be effective drug candidates. These models can filter out less promising candidates early in the drug development process,

saving resources and time (Santagati et al., 2023). A practical use-case of this application is represented by material modeling (Lourenço et al., 2023), which can be efficiently implemented with QML.

## 7.2 Quantum Sensing

Quantum Sensing is an advanced sensor technology that detects changes in motion and electromagnetic fields by collecting data at the atomic level (Degen et al., 2017).

GPS, radar, lidar, and other electromagnetic technologies use quantum physics to provide increasingly common tools on city streets. In aircraft and even on basic cell phones, quantum sensing is in the process of shifting from being a highly-prized capability few can afford to being in daily use everywhere. Once it achieves widespread adoption, quantum sensing is expected to improve capabilities for the following domains dramatically:

- Aircraft, Automobile and Electronics Manufacturers

- Geology and Civil Engineering

- Environmental Management

- Weather Forecasting

- Cosmic wave detection and simulations (gravitational wave sensing)

As witnessed by the increasing number of startups in this field, quantum sensors are finding their way from laboratories to the real world. The atomic length scale of quantum sensors and their coherence properties enable unprecedented spatial resolution and sensitivity. Biomedical applications ranging from brain imaging to single-cell spectroscopy (Aslam et al., 2023) could benefit from these quantum technologies, but evaluating the potential impact of the techniques is not trivial.

Several projects in the field of civil engineering have launched the adoption of quantum sensing technologies. Since quantum sensors allow more precise and accurate measurements, they relieve and facilitate in making the right decisions for critical tasks (e.g., underground detection of sinkholes, which currently are manually done by experts using their personal knowledge with minimum technological support. This causes a massive amount of resources and time wasted due to the lack of accuracy in correctly identifying the areas where to dig holes.

Metrology is another applicable domain for quantum sensing (Ohshima, 2022). The current interests in this direction demand compact and susceptible measurement systems. Integrating electronics with quantum sensing technology should be one of the viable areas of interest for NISQ devices.

## 7.3 Quantum Cryptography

This application domain entirely focuses on creating new and improved cryptography schemes that ensure security against attacks and threats by quantum breach technology. Proactively taking measures against the utilization of quantum technology for breaking such schemes is referred to as Quantum Safe cryptography. Keeping that in mind, currently used encryption methods for many critical communication and transaction processes use the Rivest–Shamir–Adleman (RSA) scheme (Rivest et al., 1978). It is fundamentally based on the keys generated from prime factors of very large numbers. Although any classical computer can find prime numbers of a given small number given the understanding of complexity theory, it has been proven that for large numbers, the complexity of finding its prime numbers becomes difficult to intractable, i.e., solvable only in indefinite time. This particular property of the problem with classical computers made it an obvious choice to use the RSA as an encryption method. Something that would require thousands of years to solve with the available technology would raise no harm. However, the works of Peter Shor (Shor, 1994; 1997) showed the capability of quantum computers to obtain solutions to problems that were presumably be indefinitely solved in the classical domain. This single instance of challenges to established security claims opens new avenues, driven by the need to develop improved and robust cryptography methods that make

the communication *Quantum Safe* (Campagna et al., 2015; Stebila & Mosca, 2016). Instead of completely changing a classical algorithm into quantum, the focus is on adding quantum modules to existing classical cryptography techniques. The advantage of this strategy is that while the existing classical techniques remain intact, applying the quantum methods improves their efficiency and speed (Bozzio et al., 2022). This proved to be a very useful domain of potential applications for QC that builds over the existing schemes and makes them robust and efficient rather than re-inventing the whole process.

## 7.4 Financial Modeling

Quantum-enhance portfolio optimization and risk analysis have been demonstrated on small benchmark problems, showing that variational quantum algorithms can achieve comparable solutions to classical solvers with fewer iterations (Orús et al., 2019; Egger et al., 2020). For instance, the VQE and QAOA have been applied to relatively small portfolio instances, yielding near-optima allocations in simulation (Havlíček et al., 2019; Zaman et al., 2024b). Likewise, quantum-inspire Monte Carlo methods leverage amplitude estimation to compute value at risk and option pricing with quadratic speedup in the fault-tolerant regime (Montanaro, 2015; Woerner & Egger, 2019). Moreover, hybrid quantum-classical ML systems have been proven successful for financial fraud detection prevention (Grossi et al., 2022; Innan et al., 2024b).

## 7.5 Logistics Optimization

Supply chain and logistics professionals have been overloaded over the past several years. An increasing amount of uncertainties, extreme weather conditions, and pandemic-fueled supply and demand changes have exponentially increased logistics complexity. Combinatorial optimization in routing and scheduling can be framed as QUBO problems and solved via quantum annealing or QAOA. Early studies mapped small Vehicle Routing Problems (VRPs) to D-Wave hardware, achieving feasible tours but with limited scaling due to embedding overhead (Venturelli et al., 2015; Neukart et al., 2017). Variational algorithms on gate-model devices have tackled small Traveling Salesman Problem (TSP) instances, demonstrating proof-of-concept quantum speedups (Cerezo et al., 2021a). In practice, embedding real-worl logistics networks (hundreds of nodes) remains out of reach on NISQ devices. Hence, hybrid workflows partition the problem into quantum-solvabl subgraphs and classical refinement loops (Chicano et al., 2025).

## 7.6 Environmental, Social, and Governance (ESG)

Quantum sensing for environmental monitoring exploits enhanced sensitivity of entangled states. Proof-of-principle experiments using spin-squeezed Bose–Einstein condensates have measured magnetic fields with precision beyond the standard quantum limit (Pezze et al., 2018; Hosten et al., 2016). Such sensors could track pollutant concentrations or micro-climat variables with greater accuracy, benefiting ESG reporting. QML can support ESG goals by optimizing renewable-microgri scheduling and emissions reduction. For instance, a hybrid quantum-classica QAOA approach to microgrid economic dispatch reported lower operational cost and resilience under fluctuating renewable supply compared to classical methods (Koretsky et al., 2021). In environmental monitoring, quantum-enhance sensing algorithms using entangled probes have demonstrated improved precision in detecting trace gases (Degen et al., 2017). However, broad ESG impact hinges on scaling these methods beyond prototypical demos toward fault-toleran platforms and integrating them into existing infrastructure.

# 8 Conclusion and Road Ahead

The development of the QC field, particularly in QML, opens new avenues for demonstrating the true potential of technological advancements in various subfields. Despite the rapid and substantial progress, QML faces significant challenges related to the size and noise constraints, which affect the *scalability* and *reliability* of these systems. The path towards FTQC must be pursued diligently, following the steps outlined the roadmap for QC development. To achieve groundbreaking advancements in this relatively new and exciting field, it is essential to adopt innovative approaches rather than strictly adhering to traditional research and development steps. For QML implementations, finding analogies between classical ML and

QML is not always necessary. Instead, exploring alternative perspectives to redefine metrics and standards from a quantum viewpoint is crucial. The open research directions can be summarized as follows.

- **Benchmarks:** Establishing benchmark suites for QML is essential for creating a standardized framework for evaluating and comparing different QML algorithms and infrastructures. The current lack of comprehensive comparative frameworks and standardized hybrid QML implementations hampers progress. In classical ML, benchmarks and state-of-the-art algorithms are well-established, providing clear metrics for performance evaluation. Although QML algorithms are still in their early stages and may not yet match the accuracy of mature classical ML algorithms, quantum computing offers unique opportunities to redefine comparison metrics and develop new benchmarks that capture the strengths of QML.

- **Scalability:** Achieving scalability in quantum computing involves moving beyond single quantum chips to develop multi-chip devices with efficient interconnection and communication systems. Companies like IBM and Honeywell are making strides toward large-scale quantum devices, as shown in Figure 14, but significant challenges remain. The primary focus should be on minimizing error rates and enhancing system reliability. As suggested in (Preskill, 2018), reducing error rates is a top priority for making quantum computation technology both impactful and widely adopted. Innovations in error correction and fault-tolerant quantum computing will be crucial for scaling up quantum systems.

- **Stable Technology Standardization:** To suppress noise and enable computations in noisy environments, it is vital to establish protocols that effectively characterize noise correlations between qubits in a scalable and cost-effective manner. The advanced technology required for quantum computing can be expensive, necessitating a focus on developing fault-tolerant systems and cost-efficient error-reduction methods. Balancing cost and reliability is key to selecting suitable technologies. Efforts should be directed toward designing robust error correction techniques that maintain performance while keeping costs manageable.

- **NISQ Limitations:** In the NISQ era, where quantum devices have limited qubits and high noise levels, it is crucial to design optimizations tailored to these constraints. Realistic expectations for QML technology deliverables are necessary, recognizing that while quantum computers have the potential to outperform classical systems, current hardware limitations must be accounted for. Developing QML algorithms and optimization strategies that operate efficiently within the context of these limitations is essential. This includes leveraging hybrid quantum-classical approaches, where classical preprocessing and postprocessing can complement quantum computations.

Finally, it is important to explicitly acknowledge broader ethical and societal impacts associated with QC and QML. The emergence of large-scale fault-tolerant quantum computers poses substantial threats to current cryptographic standards (e.g., RSA encryption), potentially undermining cybersecurity infrastructures and necessitating timely development and adoption of quantum-safe cryptographic methods. Moreover, access to quantum computing resources and expertise remains limited and costly, potentially exacerbating existing inequalities between institutions and nations with differing economic capabilities. The significant energy demands of current quantum systems, especially those reliant on cryogenic cooling, further highlight environmental sustainability concerns that must be addressed as this technology scales. Additionally, QML inherits ethical considerations common to classical AI/ML, such as data biases, fairness issues, transparency, interpretability challenges, and risks of misuse in surveillance or autonomous weapon systems. Addressing these ethical and societal challenges proactively will be essential for responsible and equitable advancement in quantum technologies.

This survey provides a comprehensive overview of the current state-of-the-art algorithms, datasets, tools, and applications of QML. It serves as a reference resource to instruct and motivate passionate readers to learn and contribute to the research and design of innovative methodologies, tools, algorithms, and technologies that facilitate advancements in this emerging field. By exploring these directions, we can push the boundaries of what is possible with QML and pave the way for its broader adoption and impact.

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

## A    Complexity Theory

This section provides an overview of complexity classes. A *promise problem* is a decision problem where the input is to be selected from all possible input strings. A problem with an explicitly defined structure or characteristics of input is defined in advance, making the problem environment-entrusted with certain types of promised input with no instance of input sent in unexpectedly or out of definition. All promise problems are assigned to a complexity class suitable to their nature and requirements. Formally, *complexity classes* are a group of computational problems that have similar resource-based complexity (Babai et al., 1986). In classical computing, there exist the following classes:

- *Polynomial Class(P)*: It contains all decision problems that can be solved by a deterministic Turing machine using a polynomial amount of computation time, or polynomial time.

- *Non-deterministic Polynomial (NP)*: It is the collection of decision problems that can be solved by a non-deterministic machine in polynomial time but these problems of NP can be verified by a Turing machine in polynomial time.

- *NP-hard*: An NP-hard problem is at least as hard as the hardest problem in NP and it is a class of problems such that every problem in NP reduces to NP-hard. Since it takes a long time to verify their solution, all NP-hard problems are not in NP.

- *NP-complete*: A problem is NP-complete if it is both NP and NP-hard. NP-complete problems are the hardest problems in the NP space. If one could solve an NP-complete problem in polynomial time, then one could also solve any NP problem in polynomial time.

## B    Quantum Technical Concepts and Definitions

### B.1    Hilbert Space

In Quantum mechanics the state of a physical system is represented by a vector in the Hilbert space, which is a complex vector space with an inner product. In the finite-dimensional complex vector spaces that come up in quantum computation and quantum information, the Hilbert space is exactly the same thing as an inner product space (both terms can be used interchangeably)[2].

### B.2    CHSH Inequality

The Clauser–Horne–Shimony–Holt (CHSH) inequality is a key result in quantum physics that distinguishes the predictions of quantum mechanics from classical physics, particularly those based on local hidden variable theories. In a CHSH experiment, two observers (Alice and Bob) make measurements on entangled particles using randomly chosen settings (Bell, 1964). Classical theories constrained by locality and realism predict a specific bound on the correlation between their outcomes, encapsulated by the CHSH inequality(Clauser et al., 1969).

Quantum mechanics, however, predicts stronger correlations that violate this classical bound when entangled particles are involved (Cirel'son, 1980). Repeated experiments have confirmed this violation, supporting quantum mechanics and challenging classical assumptions (Aspect et al., 1982; Hensen et al., 2015; Shalm et al., 2015; Giustina et al., 2015). These findings not only deepen our understanding of quantum entanglement but also form the basis for future quantum computing technologies.

### B.3    Wave-Particle Duality

The *Wave-Particle duality* theory posits that waves can exhibit particle-like properties and particles can exhibit wave-like properties. This duality is a cornerstone of quantum mechanics, contrasting sharply with

---

[2]For infinite dimensions, the Hilbert space satisfies additional technical restrictions above and beyond inner product spaces (Nielsen & Chuang, 2010b).

the principles of classical mechanics or Newtonian physics (Wheaton, 2009). In classical mechanics, waves and particles are distinct entities with no overlap in their properties. Quantum mechanics, however, reveals that entities such as photons and electrons can behave both as particles and as waves, depending on the experimental setup. This dual nature is exemplified in phenomena such as the double-slit experiment, where particles like electrons create an interference pattern characteristic of waves when not observed, but act like particles when observed.

## B.4 Max-Born Rule

The *Max-Born rule* provides a probabilistic framework for quantum mechanics. It states that the probability density of finding a quantum system in a given state is proportional to the square of the amplitude of the system's wave function at that state. This is usually represented within a state-vector formalism in Hilbert space. The rule is fundamental to the measurement process in quantum mechanics (Stoica, 2023). When a qubit is measured, the probability of finding it in a particular state is determined by the square of the amplitude of its wave function for that state.

## B.5 State Space and State Vector

The Hilbert space is associated with any isolated physical system, known as the state space of the system (Nielsen & Chuang, 2010b). The system is completely described by its state vector, which is a unit vector in the system's state space.

## B.6 Schrodinger's Equation

The Schrödinger's equation (see Equation (3)) describes the evolution of the state $\psi$ of a closed quantum system over time $t$ (Nielsen & Chuang, 2010b).

$$i\hbar\frac{\partial\psi}{\partial t} = \hat{H}\psi \tag{3}$$

## B.7 Hermitian Matrix/Operator

In the Schrödinger's equation, $\hat{H}$ is the hermitian operator, known as the Hamiltonian of the closed system. An operator $A$ whose adjoint is $A$ is known as a Hermitian or self-adjoint operator. It is a complex square matrix that is equal to its own conjugate transpose.

### B.7.1 Hamiltonian

In principle, if we know the Hamiltonian of a system, together with the Planck's constant ($\hbar$) in the Schröndinger's equation, we can understand the complete dynamics of the physical system. In reality, determining the Hamiltonian of a physical system is a very difficult task (hard problem). Quantum computing represents a valid avenue toward realizing such physical systems with computations closer to real systems. In quantum computation and information, we usually consider the Hamiltonian as a starting point. Since the Hamiltonian is a Hermitian operator, it has spectral decomposition with eigenvalues corresponding to the normalized eigenvectors. The states $|E\rangle$ are conventionally referred to as energy eigenstates or sometimes stationed states and $E$ is the energy of the state. The lowest energy state is the ground energy with the corresponding eigenstate known as the ground state.

## B.8 Observable

Projective measurements are a special case of general measurements that have the ability to perform unitary transformations. A projective measurement is described by an observable ($M$) and a Hermitian operator on the state space of the system being observed. The observable has a spectral decomposition. The completeness relation, whose probabilities sum to 1, is applicable to projective measurements and makes the

probability constraint valid. This way of computing measurements is related to the Heisenberg uncertainty principle (Nielsen & Chuang, 2010b).

### B.9 Heisenberg's Uncertainty Principle

Quantum noise arises from imperfections in quantum operations and the measurement process, rather than from inherent randomness in quantum mechanics itself. Although quantum dynamics are fundamentally deterministic, practical quantum systems are subject to environmental interactions and operational inaccuracies, causing deviations from ideal quantum states (Nielsen & Chuang, 2010b). While the Heisenberg uncertainty principle limits the simultaneous precision of certain observables, it does not directly produce noise. In contrast, quantum noise is primarily due to errors introduced during state preparation, gate operations, and measurements, collectively impacting system fidelity and performance (Preskill, 2018).

Heisenberg's uncertainty principle states that it is fundamentally impossible to simultaneously measure the exact value of certain pairs of related physical quantities, such as position ($x$) and momentum ($p$), with infinite precision (Heisenberg, 1925). Mathematically, this principle is expressed as in Equation (4):

$$\Delta x \cdot \Delta p \geq \frac{\hbar}{2} \tag{4}$$

where $\Delta x$ is the uncertainty in position, $\Delta p$ is the uncertainty in momentum, and $\hbar$ is the reduced Planck's constant. This principle highlights the intrinsic limitations in our ability to measure and predict the behavior of quantum particles accurately.

Heisenberg's uncertainty principle provides a fundamental explanation for the presence of quantum noise, which arises from the inherent uncertainties in quantum systems. Qubits, as quantum mechanical entities, are particularly vulnerable to this noise. Addressing these challenges is essential for advancing quantum computing and harnessing its full potential.

