# OpenReview forum: "A Survey on Quantum Machine Learning: Basics, Current Trends, Challenges, Opportunities, and the Road Ahead"
_TMLR — Rejected by TMLR_

### Review · Reviewer_YDco · 2025-04-15

**Summary Of Contributions:**

This work introduces a wholistic review of quantum computing and quantum machine learning, providing a review that engages with all levels of the quantum stack.

**Audience:**

Yes

**Broader Impact Concerns:**

This reviewer does not have broader ethical concerns for this work.

**Claims And Evidence:**

Yes

**Requested Changes:**

Major fixes:
- It's hard for me to give a specific list of proposed adjustments that are critical to secure my recommendation. I think that list would be very large. In summary, addressing the above cons points. Going through every section, making it clear what the point is, explaining the points for non-experts or even people who don't know QC and synthesizing the citations into a narrative that is coherent. The connecting all of these sections into a structured paper.

Minor fixes:
- Figure 3 beta^2 is messed up

**Strengths And Weaknesses:**

Pros:
- I like the style of many of the figures

Cons:
- This paper has a large number of substantial hurdles to overcome. In general, I would summarize the main issue (that comes across in almost every section) being that the paper doesn't function well as a review paper. Basically every section is very short with just a one sentence introduction, with no meaningful synthesis or explanation, then maybe 1-2 citations. This isn't useful for people knowledgable in the field, because it lacks the citations for people to work with (and also doesn't synthesize any information about these citations in a meaningful way). And it isn't useful to new comers because the information is not explained at all.
- Given the target audience of non-QML experts, there is a lot of pedagogical work missing. Ideas get brought up with 0 explanation, simply called forth from the void (maybe with a citation), then dropped and never discussed again (there are so many examples, I won't list them all, as a random example "re-uploading" it only shows up once, is never explained than is never brought up again, same with Bloch sphere, etc.).
- There is no central narrative or coherent flow to much of the paper
- There is much to be desired from a writing perspective. Sentences don't always make total sense (e.g. “QC has the potential to solve problems more cost-efficiently through optimizations, time savings, and environmentally friendly designs.” what environmentally friendly designs?, “further developed by Preskill (2018; 2023)” lots of worked further developed QC between those decades, “Unlike classical systems that struggle to encompass natural processes” encompass doesn't really make sense here, “a qubit can exist in a superposition of both 0 and 1 simultaneously. This means that a qubit can exist in a superposition of both 0 and 1 states simultaneously” repeats etc.). Normally I wouldn't make a hassle over nitpicking sentences, but it's so pervasive and ubiquitous that it has a noticeably negative impact on the reader.
- The choice of citations doesn't always make sense. Even in the introduction, no notable QML/ML works are cited, just a collection of minimally cited works that aren't connected to the paper. In other sections, the core papers beyond that idea aren't cited (e.g. reuploading)
- Not including the foundational math is actually a decision I am okay with, but math seems to be randomly interjected with no explanation (e.g. a basis state |0> is never even explained).
- Lots of ideas that aren't really related to QC/QML come up (like really wave-particle duality doesn't need to be discussed)
- No discussion of density matrices for some reason. Mixed and pure states come up, but with no explanation or mathematical framework.
- Figure 7 is totally unclear if you don’t know what you are looking at, people have never seen "gates" before, or what wires are, nothing is explained
- Entanglement is one of the most important (and classically misunderstood) aspects of QC, and warrants more attention (CHSH inequality, etc.)
- Quantum noise section makes very little sense, nothing about kraus operators or density matrices or if you wanted to avoid math any discussion of noise channels and how noise impacts/works within a system. Also no citations either. The uncertainty principle also adds nothing there. I'm calling out specific sections, but really this ties into the points I've already made. Not good citations, explanations, or narrative.
- Algorithms are discussed in terms like “in a very fast manner” that is not helpful for computation
- The complexity section is misleading. Things like “quantum supremacy” (which I will note, is now not just not supremacy but it is now slower to do that task on QPUs than GPUs https://arxiv.org/abs/2406.18889) did not cause people to re-evaluate complexity theory, the complexity of tasks such as random circuit sampling is what led companies to use it as a benchmark. It also offers no meaningful explanation but I’m just beating a dead horse here
- “Due to their stability at close to normal temperatures, Majorana Qubits work fine” is definitely not how I would describe the recent developments of Majorana qubits (which may or may not even have been shown to exist https://arxiv.org/abs/2503.08944, https://www.nature.com/articles/d41586-021-00954-8, etc.)

---

> ### Author Response · Authors · 2025-05-07
> **Response to Reviewer YDco**
>
> Comment 1: Improve the paper’s function as a comprehensive and pedagogically sound review.
>
> Response:
>
> •	Significantly restructured each major section (Sections 2-7) to clearly define terms, integrate meaningful synthesis, and provide detailed explanations suitable for non-experts.
>
> •	Added clearer definitions of quantum computing basics, clarified confusing concepts like superposition, entanglement, and quantum noise.
>
> Comment 2: Improve citations and remove marginally relevant information.
>
> Response:
>
> •	Revised citation choices throughout the manuscript, focusing on highly influential foundational and contemporary works.
>
> •	Removed marginal information and provided focused, relevant context.
>
> •	Some content (e.g., wave-particle duality, uncertainty principle) has been moved to the Appendix.
>
> Comment 3: Enhance clarity and readability.
>
> Response:
>
> •	Reviewed and edited several sections of the manuscript for linguistic clarity, removing redundancies and ambiguous phrases.
>
> •	Improved sentence structure throughout.
>
> •	Figure 3 typo fixed.
>
> Comment 4: Clarify and improve the complexity and quantum supremacy section.
>
> Response:
> Revised Section 3.2 thoroughly, clarifying the complexity theory context, quantum supremacy definitions, and recent relevant findings.
>
> Comment 5: Provide clarity on the current status of Majorana qubits.
>
> Response:
> Revised Section 6.1.8, clarifying the recent developments of Majorana qubits.

---

> > ### Comment · Reviewer_YDco · 2025-05-16
> >
> > The described changes are in a positive direction and I appreciate the authors doing them.

---

### Review · Reviewer_6u5Q · 2025-04-21

**Summary Of Contributions:**

The paper provides a survey of the field of quantum machine learning. It covers some of the theoretical foundations of quantum computing and various aspects of the field such as quantum algorithms, data encodings, hardware designs, and benefits.

**Audience:**

Yes

**Broader Impact Concerns:**

N.A.

**Claims And Evidence:**

No

**Requested Changes:**

The following list of suggested changes is incomplete, but hopefully provides some help to improve the paper.

- Last sentence of the abstract: This is ideally written in a review of the survey, but maybe not in the paper itself.
- Regarding Moore's law: Please provide a more recent reference, there is an ongoing debate about Moore's law.
- The structure and purpose of Section 2 is unclear to me. The explanations of qubits in 2.1, 2.2.1, and 2.2.2 are very wordy but do not contain a proper definition. It is also in general questionable whether a survey needs to be accessible to readers not familiar with elementary quantum mechanics. Many concepts are mentioned but cannot be understood without knowing them anyway, and it is not clear why they are relevant for the survey (Bloch sphere, wave particle duality, Max-Born rule). Several statements are questionable (see following points)
- The last paragraph on page 6 is misleading: Quantum mechanics is not just about randomness, note also that thermodynamics covers randomness without necessarily being quantum mechanical.
- Quantum gates and quantum circuits are crucial for QML and should probably be explained more carefully (if the survey should be that broad). Why unitary, what types of gates.
- 'These eigenvalues are the result of the system being represented by a Hermitian matrix' What does this mean?
- Section 2.5.1: Why is this mentioned again after discussing 2 qubit systems before.
- 'The inherent uncertainty in quantum mechanics leads to quantum noise, which manifests as fluctuations in the...': This is not true. Quantum mechanics is perfectly deterministic, only the measurement introduces randomness. It is just difficult to implement the desired operations with sufficiently high fidelity.
- How do Sections 2.6.2-2.7. fit into the goal of Section 2
- While many aspects are covered that are not closely related to QML, the measurement aspect of quantum mechanics is hardly discussed even though this is a crucial ingredient. It is somehow alluded to in Section 2.5 but never pinpointed.
- My overall recommendation would be to mostly remove Section 2 because many standard textbooks provide an introduction to QM
- The structure of Section 3 is also unclear. Why is dimensionality reduction mentioned, there are many further quantum algorithms? Sections 3.3 and 3.5 are very superficial, why is the goal of Section 3.4 relevant? Also this needs to be seen in light of a measurement that destroys the quantum state. An alternative to structure this section would be to start from the complexity theoretic viewpoint, then outline exponential speedups and then discuss polynomial speedups and their relevance in regimes with large models and a lot of data.
- The purpose of Section 4 is generally clearer, however, it is a bit unclear why this selection of methods and references are dicussed. E.g., why are convolutional networks so relevant, why is there then only a single reference in this section? What is the difference between 4.3.2. and 4.3.4?
- The Barren plateau problem should be emphasised much more: It needs to become clear that many of the suggested architectures suffer from extrem scalability problems. In addition, it should be discussed how those issues can potentially be avoided (architectures etc.).
- Section 5.1.: The list of quantum data is confusing. Generally we want to map classical data $x$ to a quantum state $\psi(x)$.
- First sentence last paragraph on page 24 is unclear. The title of 5.1 is also a bit misleading
- Maybe Section 5 can be restructured such that it is separated in classical data (preprocessing, encoding) and then quantum data (datasets, advantages of qc for quantum data). The preprocessing section is a bit lengthy as it seems to cover standard ML preprocessing (also note that classical ML often recovers features from noisy data).
- Section 6.1 was essentially covered before and there is no need to repeat this, instead it should be once explained propberly. The remaining subsections are a bit superficial.
- 7.4-7.6. are a bit speculative.
- Several sections of the survey were flagged as AI generated by an AI detector. Please clarify how/if LLMs were used for the paper.

**Strengths And Weaknesses:**

# Strengths

- The paper covers many aspects relevant to the field of quantum machine learning.
- It is highly appreciated that many figures are included.
- The paper tries to convey intuitions.

# Weaknesses

In my opinion, the two main weaknesses are as follows:
- I am not convinced that the paper achieves the goals outlined in the acceptance criteria for surveys (draw new connections, highlight trends, and suggest new problems in an area), in the current form. The survey mostly seems to be a list of results relevant to the area but does not provide sufficient structure to understand their significance.

- There are several parts of the review (mostly in the foundations part) which appear at least misleading (see below).

Therefore, I think that the structure of the paper needs to be improved, and it needs to become clearer what additional value this survey brings to the field (according to Table 1, more than 10 surveys appeared in the last 3 years). Just covering more aspects more superficially than prior work is not sufficient. In addition, many parts can be shortened substantially, there are many sentences containing only marginal relevant information. On the other hand, many concepts are explained extremely terse which probably makes it impossible to even understand the gist if one is not familiar with them anyways. Also, a more formal writing could help to avoid misunderstandings.

---

> ### Author Response · Authors · 2025-05-07
> **Response to Reviewer 6u5Q**
>
> Comment 1: Last sentence of the abstract: This is ideally written in a review of the survey, but maybe not in the paper itself.
>
> Response:
> The last sentence of the abstract has been revised to summarize the main aim of this survey.
>
> Comment 2: Regarding Moore's law: Please provide a more recent reference, there is an ongoing debate about Moore's law.
>
> Response:
> Section 1 has been revised to provide an overview of the debate about Moore’s law.
>
> Comment 3: Clarify and restructure Section 2 (Quantum Computing Preliminaries).
>
> Response:
>
> •	Section 2 has been substantially rewritten to remove redundant information, clarify definitions, and some content has been moved to the Appendix.
>
> •	Removed overly verbose explanations and added formal definitions of key concepts.
>
> •	Revised misleading statements regarding quantum randomness, clearly distinguishing measurement-induced randomness from deterministic quantum evolution.
>
> •	Quantum gates, quantum circuits, and measurement aspect of quantum mechanics explained more carefully.
>
> •	Subsections revised to fit into the goal of Section 2.
>
> Comment 4: Improve structure and clarity of Section 3 (Quantum Advantages).
>
> Response:
>
> •	Section 3 reorganized with clearer logical flow, removing superficial discussions and emphasizing complexity-theoretic viewpoints.
>
> •	Expanded explanations of quantum supremacy and its implications.
>
> Comment 5: Explain the selection of methods and references in Section 4. Highlight and expand on barren plateau problem.
>
> Response:
>
> •	The selection of QML methods discussed in Section 4 focuses on algorithms widely recognized in the quantum machine learning literature for their relevance in near-term quantum computing contexts, particularly hybrid quantum-classical models. Convolutional architectures such as QCNNs and Quanvolutional networks are emphasized due to their demonstrated effectiveness in practical image-based tasks and their suitability for implementation on current NISQ hardware. Additional references have been incorporated to support the significance of these approaches.
>
> •	The item on the barren plateau problem has been significantly expanded in Section 4.8. It now explicitly emphasizes the fundamental training challenges posed by barren plateaus, particularly in scaling variational quantum algorithms like VQEs, QCNNs, and quantum kernel methods. Moreover, we discuss potential mitigation strategies to address barren plateaus. This expanded discussion clarifies the practical implications of barren plateaus and highlights avenues for mitigating their impact on QML scalability.
>
> Comment 6: Clarify and restructure Section 5 (Quantum Data).
>
> Response:
> Section 5 was reorganized, clearly separating classical data preprocessing from quantum data encoding, with improved context on dataset usage.
>
> Comment 7: The information in Section 6.1 is redundant and the content in the other subsections is superficial.
>
> Response:
>
> •	Subsection 6.1 has been removed to avoid redundancy.
>
> •	More information about pros and cons of leading quantum technologies have been provided at the end of Section 6.
>
> Comment 8: Sections 7.4-7.6. are a bit speculative.
>
> Response:
> Subsections 7.4-7.6 have been revised to reduce the speculative tone and relevant techniques have been discussed.
>
> Comment 9: Several sections of the survey were flagged as AI generated by an AI detector. Please clarify how/if LLMs were used for the paper.
>
> Response:
> We acknowledge the reviewer’s concern. We clarify that no sections of the survey were generated using generative AI tools for creating new technical content or literature synthesis. However, we did make sporadic use of language models (e.g., ChatGPT) solely to assist in rephrasing or improving the clarity and grammar of content that we had originally written. All technical formulations, citations, analyses, and conclusions were composed by the authors based on a thorough literature review and domain expertise. We ensured that the use of such tools did not compromise the originality or accuracy of the material presented.

---

> > ### Comment · Reviewer_6u5Q · 2025-05-23
> >
> > I thank the authors for their updated version. The updates improve the paper.

---

### Review · Reviewer_YXua · 2025-04-23

**Summary Of Contributions:**

This paper presents a survey of Quantum Machine Learning (QML). It begins by establishing foundational Quantum Computing (QC) concepts, including qubits, superposition, entanglement, noise considerations (NISQ, error correction/mitigation, FTQC), and the quantum computing stack. The paper discusses potential QC advantages over classical methods before going into a wide review of QML algorithms. This includes variational approaches (VQAs, PQCs), quantum optimization techniques (QAOA, VQE, QA), various Quantum Neural Network architectures (QNNs, QCNNs, QGANs, etc.), Quantum Kernels, and Bayesian methods, alongside a discussion of current limitations.

A significant portion focuses on practical aspects: quantum data representation, numerous data encoding techniques, a list of relevant datasets, an overview of competing qubit hardware technologies (superconducting, trapped ion, photonic, etc.), and a comprehensive guide to software tools (simulators, libraries like Qiskit, PennyLane, Cirq) and cloud platforms. The survey concludes by exploring potential QML applications in domains like drug discovery, finance, cryptography, and logistics, and compares its scope against previous QML surveys. The overall aim is to provide a comprehensive resource covering the QML landscape from fundamentals to practical implementation and future directions.

**Audience:**

Yes

**Broader Impact Concerns:**

The paper itself, being a survey, does not introduce new ethical concerns directly. However, the field of QML and QC, which it covers, has potential broader impacts that warrant consideration:

1. As mentioned in Section 7.3, large-scale fault-tolerant QC poses a significant threat to current public-key cryptography standards (like RSA). While the paper discusses quantum-safe cryptography as a response, the societal implications of breaking current encryption are substantial.
2. Access to quantum computing hardware and expertise is currently limited and expensive, potentially exacerbating inequalities between well-funded institutions/nations and others.
3. While QC might solve certain problems more efficiently, the energy requirements for operating current (especially cryogenic) quantum computers are significant. The overall environmental impact as the technology scales needs consideration, although the paper touches upon ESG potential positively.
4. QML inherits many ethical concerns from classical AI/ML, such as potential biases encoded in data (even if processed on a QC), fairness, transparency (interpretability of QML models), and potential misuse in areas like surveillance or autonomous weapons if performance surpasses classical methods.

The authors could optionally add a brief statement acknowledging these potential impacts, particularly regarding cryptography and resource accessibility, to provide a more complete picture.

**Claims And Evidence:**

Yes

**Requested Changes:**

1. For a few central algorithms (e.g., VQE in 4.3.3, Quantum Kernels in 4.6), consider adding a sentence or two clarifying common implementation choices (e.g., types of Ansatze for VQE, the role of feature maps for kernels) or associated practical challenges (e.g., optimization difficulties, classical simulability).
2. In sections discussing multiple related approaches (e.g., Section 4.4 on QNN variants, Section 6.2 on hardware, Section 6.4 on software), consider adding brief comparative statements.
2. In sections discussing specific algorithms (Section 4) and applications (Section 7), consider acknowledging the current reliance on future hardware improvements or specific conditions (e.g., classically hard feature maps) needed for demonstrating clear advantages over optimized classical algorithms.
3. Consider adding a sentence in relevant sections (e.g., 2.6.2 NISQ Era, 4.8 Limitations) that provides illustrative context on the gap between current hardware capabilities (e.g., qubit counts, coherence times) and the potential resource requirements for achieving practical advantage with certain QML algorithms on large-scale problems.

**Strengths And Weaknesses:**

**Strengths:**

The survey covers a wide range of topics within QC and QML, from fundamental physics principles to specific algorithms, hardware implementations, software tools, datasets, and applications. The claim of broad coverage, supported by Table 1, appears justified.

The paper follows a logical progression, starting with QC basics, moving to QML algorithms and practical considerations (data, tools), and concluding with applications and future outlook. This structure makes it accessible for readers with varying levels of prior knowledge.

Concepts are generally explained clearly, with figures (e.g., Bloch sphere, superposition examples, circuit diagrams, hardware types) and tables (comparison of surveys, datasets, hardware, simulators) to aid understanding. Foundational concepts like superposition and entanglement are introduced before discussing their use in QML.

Strong emphasis is placed on the practical aspects of QML, including detailed sections on available hardware platforms (Table 3), software libraries and simulators (Table 4), data encoding methods, and cloud platforms. This is highly valuable for researchers and engineers looking to engage with QML development.

The survey acknowledges the limitations and challenges of current QML (Section 4.8) and the NISQ era (Section 2.6.2, Section 8), including noise, scalability issues, barren plateaus, and the gap between theoretical potential and demonstrated advantage. It also includes some critical discussion regarding "quantum enhancement" versus proven speedups (Section 3.3).

**Weaknesses:**

While covering a broad range of algorithms, the description of individual methods (e.g., VQE, Quantum Kernels) sometimes lacks sufficient detail regarding common variants, implementation choices, or specific associated challenges (like Ansatz selection or feature map design).

While the survey covers many items within categories (e.g., different QNN types, hardware technologies, software tools), it rarely provides direct comparisons between them based on relevant metrics or use cases. For example, the pros and cons of QCNNs vs. QuanCNNs for specific tasks, or the trade-offs between superconducting and trapped-ion qubits specifically for running variational algorithms, are not explicitly discussed.

While Table 2 lists datasets, the survey provides limited context on how these datasets are typically used in QML research, the specific challenges associated with encoding them (beyond the general encoding techniques listed), or common benchmark tasks performed on them. This makes it harder for readers to select appropriate datasets for their own experiments.

Section 6.4 lists numerous software frameworks (Qiskit, PennyLane, Cirq, etc.) but doesn't sufficiently differentiate their core philosophies, primary strengths, or typical target use cases (e.g., PennyLane's focus on differentiable programming vs. Qiskit's extensive hardware backend integration).

The discussion of quantum advantage, particularly in the applications section (Section 7), could benefit from more consistently integrating the caveats associated with NISQ limitations and the ongoing research needed to demonstrate practical, end-to-end speedups over state-of-the-art classical methods for specific problems.

While NISQ limitations are discussed conceptually, providing some indicative quantitative context (e.g., typical qubit counts/coherence times versus potential algorithmic requirements) could further help readers appreciate the scale of current hardware challenges relative to QML algorithm demands.

---

> ### Author Response · Authors · 2025-05-07
> **Response to Reviewer YXua**
>
> Comment 1: Add details and implementation choices to algorithms such as VQE and Quantum Kernels (Section 4.3.3 and 4.6).
>
> Response:
>
> •	Section 4.3.3 (VQE) was expanded to include common ansatz types and optimization challenges.
>
> •	Section 4.6 (Quantum Kernels) now explicitly discusses feature maps and classical simulation challenges.
>
> Comment 2: Provide comparative statements between related approaches in Sections 4.4 (QNN variants), 6.2 (hardware), and 6.4 (software).
>
> Response:
> Comparative summaries highlighting pros and cons have been added explicitly to Sections 4.4, 6.2, and 6.4.
>
> Comment 3: Address current reliance on future hardware improvements and classically hard feature maps explicitly in algorithms (Section 4) and applications (Section 7).
>
> Response:
> Explicit clarifications have been included in Sections 4 and 7 regarding reliance on future hardware developments and conditions necessary for quantum advantage.
>
> Comment 4: Provide illustrative context about current hardware capabilities and resource requirements (Sections 2.6.2 NISQ Era, 4.8 Limitations).
>
> Response:
> Sections 2.6.2 and 4.8 now include quantitative metrics such as typical qubit counts and coherence times.
>
> Comment 5: Provide additional context on dataset usage in Table 2, including encoding challenges and common benchmark tasks.
> Response:
> Section 5 has been expanded with additional context on typical QML research use-cases for each dataset reported in Table 2, specific encoding challenges, and commonly employed benchmark tasks. A paragraph summarizing this context has also been added to Section 5.
>
> Broader Impact Concerns:
>
> Comment: Explicitly acknowledge broader ethical impacts of QML and QC, including threats to cryptography, access inequalities, environmental impact, and ethical concerns inherited from classical AI/ML.
>
> Response:
> A brief statement explicitly discussing these broader ethical concerns and impacts, particularly regarding cryptography vulnerabilities, accessibility disparities, environmental considerations, and potential ethical challenges related to biases, fairness, transparency, and misuse has been included in the conclusion (Section 8).

---

### Author Response · Authors · 2025-05-07
**General Response to Reviewers**

We sincerely thank the editor and all reviewers for their detailed and constructive feedback. In response, we have made substantial revisions throughout the manuscript to improve its clarity, completeness, and scholarly rigor. Key algorithmic sections, including VQE and Quantum Kernels (Section 4), were expanded to include implementation choices, optimization challenges, and limitations such as barren plateaus, along with relevant mitigation strategies and foundational citations. Comparative summaries were added to Sections 4.4, 6.2, and 6.4 to highlight trade-offs between QML models, quantum hardware, and software tools. The quantum data section (Section 5) was reorganized to better distinguish classical preprocessing from quantum encoding, with expanded context on dataset use cases and challenges. Sections 2 and 3 were restructured to provide a clearer narrative of quantum principles and complexity theory, and Section 3.2 was revised to more accurately reflect the implications of quantum supremacy benchmarks. We have also clarified the current hardware constraints and resource limitations in Sections 2.6.2 and 4.8. Subsections 7.4–7.6 were revised to avoid speculative claims and emphasize concrete techniques. In Section 8, we now explicitly acknowledge the broader ethical and societal implications of QML and QC, including cryptographic risks and access disparities. These revisions collectively enhance the pedagogical value, technical depth, and relevance of our survey.

---

### Decision · Action_Editor_HvWk · 2025-05-29

**Recommendation:** Reject

**Comment:**

During the review process many reviewers mention core points - most of which are adressing the questionable positioning of the paper. While the authors improved their paper based on the detailed feedback of the reviewers (this not only included smaller issues but also restructuring of sections to some degree), the major feedback requires a lot more effort to the paper, making in not mature for a publication in TMLR at this point of time. I have also read the paper and agree on this point.

It is unclear how the concerns can be addressed in the paper in its current state. Based on the issues mentioned above I recommend rejecting the submission at this time.

**Audience:**

It is not partiularily clear what kind of audience this review is made for. On the one side the authors aim at providing a comprehensive and holistic presentation of the state of the art (and the field with its background in general), making it different from previously published surveys on QML - most of which are only covering/focusing on parts of the outlined dimensions. On the other side however, as a result at many points the survey remains superficial and comes across more like a shallow introduction to the topic in general.

**Claims And Evidence:**

The reviews show a mixed picture about the paper with concerns being more prevalent. The paper claims some key contributions, i.e., investigating QML algorithms, quantum datasets, quantum hardware developments and tools, and their implications on QML. However, the paper in its current form falls short in a proper consolidation of the works and does not provide a cohesive structure as a whole. At many points (and most obviously in Section 4) the paper presents a lot of loosely coupled QML methods and algorithms (on the level of a shallow introduction) without a cohesive storyline that connects those topics. The same holds for the datasets section. The paper claims to be a survey but is more similar to an introduction, without making interesting connections within the field of QML nor to research outside this field.